# One Question Answering Model for Many Languages with Cross-lingual Dense Passage Retrieval

**Akari Asai**[†], **Xinyan Yu**[†], **Jungo Kasai**[†], **Hannaneh Hajishirzi**[†‡]

[†]University of Washington, [‡]Allen Institute for AI

{akari, xyu530, jkasai, hannaneh}@cs.washington.edu

## Abstract

We present **C**ross-lingual **O**pen-**R**etrieval **A**nswer Generation (**CORA**), the first unified many-to-many question answering (QA) model that can answer questions across many languages, even for ones without language-specific annotated data or knowledge sources. We introduce a new dense passage retrieval algorithm that is trained to retrieve documents across languages for a question. Combined with a multilingual autoregressive generation model, CORA answers directly in the target language without any translation or in-language retrieval modules as used in prior work. We propose an iterative training method that automatically extends annotated data available only in high-resource languages to low-resource ones. Our results show that CORA substantially outperforms the previous state of the art on multilingual open QA benchmarks across 26 languages, 9 of which are unseen during training. Our analyses show the significance of cross-lingual retrieval and generation in many languages, particularly under low-resource settings. Our code and trained model are publicly available at `https://github.com/AkariAsai/CORA`.

## 1  Introduction

Multilingual open question answering (QA) is the task of answering a question from a large collection of multilingual documents. Most recent progress in open QA is made for English by building a pipeline based on a dense passage retriever trained on large-scale English QA datasets to find evidence passages in English (Lee et al., 2019; Karpukhin et al., 2020), followed by a reader that extracts an answer from retrieved passages. However, extending this approach to multilingual open QA poses new challenges. Answering multilingual questions requires retrieving evidence from knowledge sources of other languages than the original question since many languages have limited reference documents or the question sometimes inquires about concepts from other cultures (Asai et al., 2021; Lin et al., 2020). Nonetheless, large-scale cross-lingual open QA training data whose questions and evidence are in different languages are not available in many of those languages.

To address these challenges, previous work in multilingual open QA (Ture and Boschee, 2016; Asai et al., 2021) translates questions into English, applies an English open QA system to answer in English, and then translates answers back to the target language. Those pipeline approaches suffer from error propagation of the machine translation component into the downstream QA, especially for low-resource languages. Moreover, they are not able to answer questions whose answers can be found in resources written in languages other than English or the target languages.

In this paper, we introduce a unified *many-to-many* QA model that can answer questions in *any* target language by retrieving evidence from *any* language and generating answers in the target language. Our method (called CORA, Fig. 1) extends the *retrieve-then-generate* approach of English open QA (Lewis et al., 2020; Izacard and Grave, 2021b) with a single cross-lingual retriever and a generator that do not rely on language-specific retrievers or machine translation modules. The multilingual

35th Conference on Neural Information Processing Systems (NeurIPS 2021).

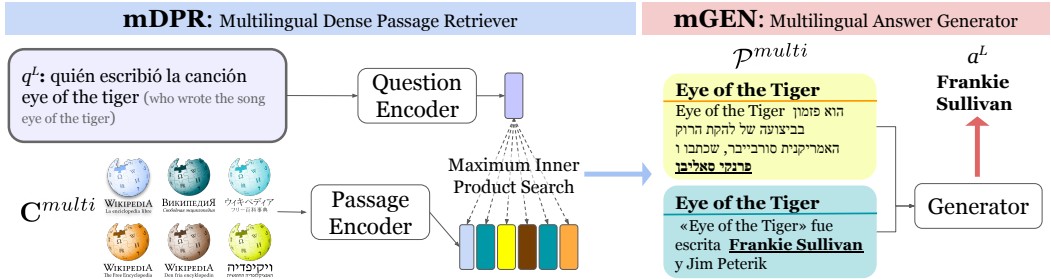

Figure 1: Overview of CORA (mDPR and mGEN).

retrieval module (**mDPR**) produces dense embeddings of a question and all multilingual passages, thereby retrieving passages across languages. The generation module (**mGEN**) is trained to output an answer in the target language conditioned on the retrieved multilingual passages. To overcome the aforementioned data scarcity issue, we automatically mine training data using external language links and train mDPR and mGEN iteratively. In particular, each iteration proceeds over two stages of updating model parameters with available training data and mining new training data *cross-lingually* by Wikipedia language links and predictions made by the models. This approach does not require any additional human annotations or machine translation, and can be applied to many new languages with low resources.

Our experiments show that CORA advances the state of the art on two multilingual open QA datasets, XOR-TYDI QA (Asai et al., 2021) and MKQA (Longpre et al., 2020), across 26 typologically diverse languages; CORA achieves gains of 23.4 and 4.7 F1 points in XOR-TYDI QA and MKQA respectively, where MKQA data is not used for training. Moreover, CORA achieves F1 scores of roughly 30 over 8 languages on MKQA that have no training data or even reference Wikipedia documents, outperforming the state-of-the-art approach by 5.4 F1 points. Our controlled experiments and human analyses illustrate the impact of many-to-many cross-lingual retrieval in improving multilingual open QA performance. We further observe that through cross-lingual retrieval, CORA can find answers to 20% of the multilingual questions that are valid but are originally annotated as *unanswerable* by humans due to the lack of evidence in the English knowledge sources.

## 2 Method

We define multilingual open QA as the task of answering a question $q^L$ in a target language $L$ given a collection of multilingual reference passages $\mathbf{C}^{multi}$, where evidence passages can be retrieved from *any language*. These passages come from Wikipedia articles that are not necessarily parallel over languages. We introduce CORA, which runs a *retrieve-then-generate* procedure to achieve this goal (Fig. 1). We further introduce a novel training scheme of iterative training with data mining (§ 2.2).

### 2.1 CORA Inference

CORA directly retrieves evidence passages from *any* language for questions asked in *any* target language, and then generates answers in the target language conditioned on those passages. More formally, the CORA inference consists of two steps of (i) retrieving passages $\mathcal{P}^{multi}$ and (ii) generating an answer $a^L$ based on the retrieved passages. $\mathcal{P}^{multi}$ can be in any language included in $\mathbf{C}^{multi}$.

$$\mathcal{P}^{multi} = \text{mDPR}(q^L, \mathbf{C}^{multi}), \ a^L = \text{mGEN}(q^L, \mathcal{P}^{multi}).$$

**Multilingual Dense Passage Retriever (mDPR).** mDPR extends Dense Passage Retriever (DPR; Karpukhin et al., 2020) to a multilingual setting. mDPR uses an iterative training approach to fine-tune a pre-trained multilingual language model (e.g., mBERT; Devlin et al., 2019) to encode passages and questions separately. Once training is done, the representations for all passages from $\mathbf{C}^{multi}$ are computed offline and stored locally. Formally, a passage encoding is obtained as follows: $\mathbf{e}_{p^L} = \text{mBERT}_p(p)$, where a passage $p$ is a fixed-length sequence of tokens from multilingual documents. At inference, mDPR independently obtains a $d$-dimensional ($d = 768$) encoding of the

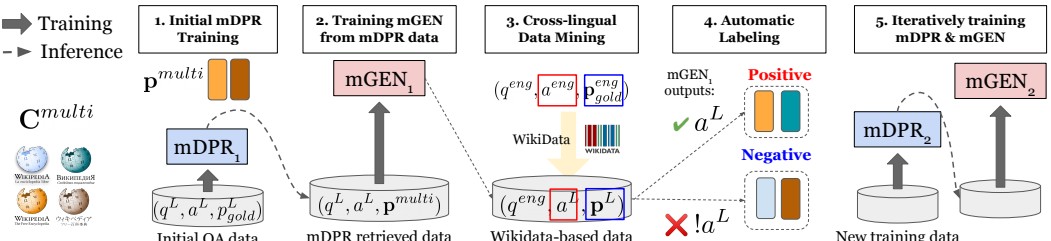

Figure 2: Overview of CORA iterative training and data mining.

question $\mathbf{e}_{q^L} = \mathrm{mBERT}_q(q^L)$. It retrieves $k$ passages with the $k$ highest relevance scores to the question, where the relevance score between a passage $p$ and a question $q^L$ is estimated by the inner product of their encoding vectors, $\langle \mathbf{e}_{q^L}, \mathbf{e}_p \rangle$.

**Multilingual Answer Generator (mGEN).** We use a multilingual sequence-to-sequence model (e.g., mT5; Xue et al., 2021) to generate answers in the target language token-by-token given the retrieved multilingual passages $\mathcal{P}^{multi}$. We choose a generation approach because it can generate an answer in the target language $L$ from passages across different languages.[1] Moreover, the generator can be adapted to unseen languages, some of which may have little or no translation training data. Specifically, the generator outputs the sequence probability for $a^L$ as follows:

$$P(a^L | q^L, \mathcal{P}^{multi}) = \prod_i^T p(a_i^L | a_{<i}^L, q^L, \mathcal{P}^{multi}), \tag{1}$$

where $a_i^L$ denotes the $i$-th token in the answer, and $T$ is the length of the answer. We append a language tag to the question to indicate the target language.

## 2.2 CORA Training

We introduce an iterative training approach that encourages cross-lingual retrieval and answer generation conditioned on multilingual passages (sketched in Fig. 2 and Alg. 1). Each iteration proceeds over two stages: **parameter updates** (§ 2.2.1) where mDPR and mGEN are trained on the current training data and **cross-lingual data mining** (§ 2.2.2) where training data are automatically expanded by Wikipedia language links and model predictions.

**Initial training data.** The initial training data is a combination of multilingual QA datasets: XOR-TYDI QA and TYDI QA (Clark et al., 2020), and an English open QA dataset (Natural Questions, Kwiatkowski et al., 2019). Each training instance from these datasets comprises a question, a positive passage, and an answer. Note that annotations in the existing QA datasets have critical limitations: positive passages are taken either from English (Asai et al., 2021) or the question's language (Clark et al., 2020). Further, most of the non-English languages are not covered. Indeed, when we only train mDPR on this initial set, it often learns to retrieve passages in the same languages or similar languages with irrelevant context or context without sufficient evidence to answer.

### 2.2.1 Parameter Updates

**mDPR updates** (line 3 in Alg. 1). Let $\mathcal{D} = \{\langle q_i^L, p_i^+, p_{i,1}^-, \cdots, p_{i,n}^- \rangle\}_{i=1}^m$ be $m$ training instances. Each instance consists of a question $q_i^L$, a passage that answers the question (positive passage) $p_i^+$, and $n$ passages that do not answer the question (negative passages) $p_{i,j}^-$. For each question, we use positive passages for the other questions in the training batch as negative passages (*in-batch negative*, Gillick et al., 2019; Karpukhin et al., 2020). mDPR is updated by minimizing the negative log likelihood of positive passages:

$$\mathcal{L}_{\mathrm{mdpr}} = -\log \frac{\exp(\langle \mathbf{e}_{q^L}, \mathbf{e}_{p_i^+} \rangle)}{\exp(\langle \mathbf{e}_{q^L}, \mathbf{e}_{p_i^+} \rangle) + \sum_{j=1}^n \exp(\langle \mathbf{e}_{q^L}, \mathbf{e}_{p_{i,j}^-} \rangle)}. \tag{2}$$

---

[1]An alternative approach of answer extraction requires translation for all language pairs (Asai et al., 2021).

**Algorithm 1:** Iterative training that automatically mines training data.

---

**Data:** Input QA pairs: $(q^L, a^L)$

1   initialize training data $\mathbf{B}^1 = (q^L, a^{\mathbf{L}}, p_{gold}), \mathbf{L} = \{\text{Eng}, L\}$;

2   **while** $t < T$ **do**

3      $\Theta^t_{mDPR} \leftarrow Train(\theta^{t-1}_{mDPR}, \mathbf{B}^t)$/* Train mDPR                      */

4      $\mathcal{P}^{multi} \leftarrow \text{mDPR}(q^L, \text{embedding}(\mathbf{C}^{multi}))$/* Retrieve passages      */

5      $\theta^t_{mGEN} \leftarrow Train(\theta^{t-1}_{mGEN}, (q^L, a^L, \mathcal{P}^{multi}))$/* Train mGEN                */

6      For $\mathbf{L}$ == Eng, $\mathcal{P}^{multi} + = \text{LangLink}(q^L, \mathbf{C}^{multi}))$ /* Mine data using Wikidata    */

7      **For** $p_i \in \mathcal{P}^{multi}$: **if** $\text{mGEN}(q^L, p_i)$ == $a^L$ **then** $positives.add(p_i)$ **else** $negatives.add(p_i)$

8      $\mathbf{B}^{t+1} += (q^L, a^L, positives, negatives)$ /* Add new training data           */

9      $t \leftarrow t + 1$

10 **end**

---

**mGEN updates** (lines 4-5 in Alg. 1). After updating mDPR, we use mDPR to retrieve top $k$ passages $\mathcal{P}^{multi}$ for each $q^L$. Given these pairs of the question and the retrieved passages $(q^L, \mathcal{P}^{multi})$ as input, mGEN is trained to generate answer $a^L$ autoregressively (Eq. (1)) and minimize the cross-entropy loss. To train the model to generate in languages not covered by the original datasets, we translate $a^L$ to other languages using Wikipedia language links and create new synthetic answers.[2] See Appendix § A.2 for more detail.

### 2.2.2   Cross-lingual Data Mining

After the parameter updates, we mine new training data using mDPR and Wikipedia language links and label the new data by mGEN predictions. This step is skipped in the final iteration.

**Mining by trained mDPR and language links** (line 4, 6 in Alg. 1). Trained mDPR can discover positive passages in another language that is not covered by the initial training data. At each iteration, we use retrieved passages $\mathcal{P}^{multi}$ for $q^L$ (line 4 in Alg. 1) as a source of new positive and negative passages. This enables expanding data between language pairs not in the original data.

To cover even more diverse languages, we use language links and find passages in other languages that potentially include sufficient evidence to answer. Wikipedia maintains article-level language links that connect articles on the same entity over languages. We use these links to expand training data from the English QA dataset of Natural Questions (line 6 in Alg. 1). Denote a training instance by $(q^{En}, a^{En}, p_{gold})$. We first translate the English answer $a^{En}$ to a target language $a^L$ using language links. We use language links again to look up the English Wikipedia article that the gold passage $p_{gold}$ comes from. We then find articles in non-English languages in the reference documents $\mathbf{C}^{multi}$ that correspond to this article. Although the language link-based automatic translation cannot handle non-entity answers (e.g., short phrases), this helps us to scale to new languages without additional human annotation or machine translation. We add all passages from these articles to $\mathcal{P}^{multi}$ as positive passage candidates, which are then passed to mGEN to evaluate whether each of them leads to $a^L$ or not.

**Automatic labeling by mGEN predictions** (lines 7-8 in Alg. 1). A passage $p_i$ from $\mathcal{P}^{multi}$ may not always provide sufficient information to answer the question $q^L$ even when it includes the answer string $a^L$. To filter out those *spurious passages* (Lin et al., 2018; Min et al., 2019), we take instances generated from the two mining methods described above, and run mGEN on each passage to predict an answer for the question. If the answer matches the correct answer $a^L$, then the passage $p_i$ is labeled as a *positive passage*; otherwise we label the input passage as a *negative passage*. We assume that when mGEN fails to generate a correct answer given the passage, the passage may not provide sufficient evidence to answer; this helps us filter out spurious passages that accidentally contain an answer string yet do not provide any clue to answer. We add these new positive and negative passages to the training data, and in the next iteration, mDPR is trained on this expanded training set (§ 2.2.1).

---

[2]This automatic answer translation is only done after the third epoch of initial training to prevent the model from overfitting to synthetic data.

# 3 Experiments

We evaluate CORA on two multilingual open QA datasets across 28 typologically diverse languages.[3] CORA achieves state-of-the-art performance across 26 languages, and greatly outperforms previous approaches that use language-specific components such as question or answer translation.

## 3.1 Datasets and Knowledge Sources

Multilingual open QA datasets differ in covered languages, annotation schemes, and target application scenarios. We evaluate F1 and EM scores over the questions with answer annotations from two datasets, following the common evaluation practice in open QA (Lee et al., 2019).

**XOR-TYDI QA.** XOR-TYDI QA (Asai et al., 2021) is a multilingual open QA dataset consisting of 7 typologically diverse languages, where questions are originally from TYDI QA (Clark et al., 2020) and posed by information-seeking native speakers. The answers are annotated by extracting spans from Wikipedia in the same language as the question (*in-language data*) or by translating English spans extracted from English Wikipedia to the target language (*cross-lingual data*). XOR-TYDI QA offers both training and evaluation data.

**MKQA.** MKQA (Longpre et al., 2020) is an evaluation dataset created by translating 10k Natural Questions (Kwiatkowski et al., 2019) to 25 target languages. The parallel data enables us to compare the models' performance across typologically diverse languages, in contrast to XOR-TYDI QA. MKQA has evaluation data only; XOR-TYDI QA and MKQA have five languages in common.

**Collection of multilingual documents $\mathbf{C}^{multi}$.** We use the February 2019 Wikipedia dumps of 13 diverse languages from all XOR-TYDI QA languages and a subset of MKQA languages.[4] We choose 13 languages to cover languages with a large number of Wikipedia articles and a variety of both Latin and non-Latin scripts. We extract plain text from Wikipedia articles using wikiextractor,[5] and split each article into 100-token segments as in DPR (Karpukhin et al., 2020). We filter out disambiguation pages that distinguish pages that share the same article title[6] as well as pages with fewer than 20 tokens, resulting in 43.6M passages. See more details in Appendix § B.2.

**Language categories.** To better understand the model performance, we categorize the languages based on their availability during our training. We call the languages with human annotated gold paragraph and answer data *seen* languages. XOR-TYDI QA provides gold passages for 7 languages. For the languages in $\mathbf{C}^{multi}$ without human-annotated passages, we mine new mDPR training data by our iterative approach (§ 2.2). We call these languages, which are seen during mDPR training, *mDPR-seen*. We also synthetically create mGEN training data as explained in § 2.2.1 by simply replacing answer entities with the corresponding ones in the target languages. The languages that are unseen by mDPR but are seen by mGEN are called *mGEN-seen*, and all other languages (i.e., included neither in mDPR nor mGEN training; 9 of the MKQA languages) *unseen languages*.

## 3.2 Baselines and Experimental Setting

We compare CORA with the following strong baselines adopted from Asai et al. (2021).

**Translate-test (MT + DPR).** As used in most previous work (e.g., Asai et al., 2021), this method translates a question to English, extracts an answer in English using DPR, and then translates the answer back to the target language. The translation models are obtained from MarianMT (Junczys-Dowmunt et al., 2018) and trained on the OPUS-MT dataset (Tiedemann, 2012).

**Monolingual baseline (BM25).** This baseline retrieves passages solely from the target language and extracts the answer from the retrieved passages. Training neural network models such as DPR is infeasible with a few thousands of training examples. Due to the lack of training data in most of

---

[3] A full list of the language families and script types are in the appendix.

[4] Downloaded from https://archive.org/details/wikimediadownloads?and%5B%5D= year%3A%222019%22.

[5] https://github.com/attardi/wikiextractor

[6] https://en.wikipedia.org/wiki/Category:Disambiguation_pages.

| Models | Target Language $L_i$ F1 | | | | | | | Macro Average | | |
|--------|------|------|------|------|------|------|------|------|------|------|
| | **Ar** | **Bn** | **Fi** | **Ja** | **Ko** | **Ru** | **Te** | **F1** | **EM** | **BLEU** |
| CORA | **59.8** | **40.4** | **42.2** | **44.5** | **27.1** | **45.9** | **44.7** | **43.5** | **33.5** | **31.1** |
| SER | 32.0 | 23.1 | 23.6 | 14.4 | 13.6 | 11.8 | 22.0 | 20.1 | 13.5 | 20.1 |
| GMT+GS | 31.5 | 19.0 | 18.3 | 8.8 | 20.1 | 19.8 | 13.6 | 18.7 | 12.1 | 16.8 |
| MT+Mono | 25.1 | 12.7 | 20.4 | 12.9 | 10.5 | 15.7 | 0.8 | 14.0 | 10.5 | 11.4 |
| MT+DPR | 7.6 | 5.9 | 16.2 | 9.0 | 5.3 | 5.5 | 0.8 | 7.2 | 3.3 | 6.3 |
| BM25 | 31.1 | 21.9 | 21.4 | 12.4 | 12.1 | 17.7 | – | – | – | – |
| Closed-book | 14.9 | 10.0 | 11.4 | 22.2 | 9.4 | 18.1 | 10.4 | 13.8 | 9.6 | 7.4 |

Table 1: Performance on XOR-FULL (test data F1 scores and macro-averaged F1, EM and BLEU scores). "GMT+GS" denotes the previous state-of-the-art model, which combines Google Custom Search in the target language and Google Translate + English DPR for cross-lingual retrieval (Asai et al., 2021). Concurrent to our work, "SER" is a state-of-the-art model, Single Encoder Retriever, submitted anonymously on July 14 to the XOR-FULL leaderboard (`https://nlp.cs.washington.edu/xorqa/`). We were not able to find a BM25 implementation that supports Telugu.

the target languages, we use a BM25-based lexical retriever implementation by Pyserini (Lin et al., 2021). We then feed the retrieved documents to a multilingual QA model to extract final answers.

**MT+Mono.** This baseline combines results from the translate-test method and the monolingual method to retrieve passages in both English and the target language. Following Asai et al. (2021), we prioritize predictions from the monolingual pipeline if they are over a certain threshold tuned on XOR-TYDI QA development set; otherwise we output predictions from the translate-test method.[7]

**Closed-book baseline.** This model uses an mT5-base[8] sequence-to-sequence model that takes a question as input and generates an answer in the target language without any retrieval at inference time (Roberts et al., 2020). This baseline assesses the models' ability to memorize and retrieve knowledge from its parameters without retrieving reference documents.

**CORA details.** For all experiments, we use a single retriever (mDPR) and a single generator (mGEN) that use the same passage embeddings. mDPR uses multilingual BERT base uncased,[9] and the generator fine-tunes mT5-base. We found that using other pre-trained language models such as mBART (Liu et al., 2020) for mGEN or XLM-R (Conneau et al., 2020) for mDPR did not improve performance and sometimes even hurt performance. We first fine-tune mDPR using gold passages from Natural Questions, and then further fine-tune it using XOR-TYDI QA and TYDI QA's gold passage data. We exclude the training questions in Natural Questions and TYDI QA that were used to create the MKQA or XOR-TYDI QA evaluation set. We run two iterations of CORA training (§ 2.2) after the initial fine-tuning. All hyperparameters are in Appendix § B.5.

# 4 Results and Analysis

## 4.1 Multilingual Open QA Results

**XOR-TYDI QA.** Table 1 reports the scores of CORA and the baselines in XOR-TYDI QA. CORA, which only uses a single retriever and a single generator, outperforms the baselines and the previous state-of-the-art model on XOR-TYDI QA by a large margin across all 7 languages. CORA achieves gains of 24.8 macro-averaged F1 points over the previous state-of-the-art method (GMT+GS), which uses external black-box APIs, and 23.4 points over the concurrent anonymous work (SER).

**MKQA.** Tables 2 and 3 report the F1 scores of CORA and the baselines on over 6.7k MKQA questions with short answer annotations[10] under *seen* and *unseen* settings. CORA significantly

---

[7]For the languages not supported by Pyserini, we always output translate-test's predictions.

[8]We did not use larger-sized variants due to our computational budget.

[9]The alternative of XLM-RoBERTa (Conneau et al., 2020) did not improve our results.

[10]Following previous work in open QA but different from the official script of MKQA (Longpre et al., 2020), we disregard the questions labeled as "no answer". As shown in our human analysis, it is difficult to prove an answer does not exist in the millions of multilingual documents even if the annotation says so.

| Setting | – | Seen (Included in XOR-TYDI QA) | | | | | | mDPR-seen | | | |
|---|---|---|---|---|---|---|---|---|---|---|---|
| | Avg. over all $L$. | **En** | **Ar** | **Fi** | **Ja** | **Ko** | **Ru** | **Es** | **Sv** | **He** | **Th** |
| CORA | **21.8** | 40.6 | 12.8 | **26.8** | **19.7** | **12.0** | **19.8** | **32.0** | **30.9** | **15.8** | **8.5** |
| MT+Mono | 14.1 | 19.3 | 6.9 | 17.5 | 9.0 | 7.0 | 10.6 | 21.3 | 20.0 | 8.9 | 8.3 |
| MT+DPR | 17.1 | **43.3** | **16.0** | 21.7 | 9.6 | 5.7 | 17.6 | 28.4 | 19.7 | 8.9 | 6.9 |
| BM25 | – | 19.4 | 5.9 | 9.9 | 9.1 | 6.9 | 8.1 | 14.7 | 10.9 | – | 4.9 |
| Closed | 4.5 | 8.0 | 4.6 | 3.6 | 6.5 | 3.8 | 4.1 | 6.6 | 4.8 | 3.8 | 2.1 |

Table 2: F1 scores on MKQA seen and mDPR-seen languages.

| Setting | mGEN-seen | | | | | | | Unseen | | | | | | | |
|---|---|---|---|---|---|---|---|---|---|---|---|---|---|---|---|
| | **Da** | **De** | **Fr** | **It** | **Nl** | **Pl** | **Pt** | **Hu** | **Vi** | **Ms** | **Km** | **No** | **Tr** | **cn** | **hk** | **tw** |
| CORA | **30.4** | **30.2** | **30.8** | **29.0** | **32.1** | **25.6** | **28.4** | **18.4** | **20.9** | **27.8** | **5.8** | **29.2** | **22.2** | **5.2** | **6.7** | **5.4** |
| MT+Mono | 19.3 | 21.6 | 21.9 | 20.9 | 21.5 | 24.6 | 19.9 | 16.5 | 15.1 | 12.6 | 1.2 | 17.4 | 16.6 | 4.9 | 3.8 | 5.1 |
| MT+DPR | 26.2 | 25.9 | 21.9 | 25.1 | 28.3 | 24.6 | 24.7 | 15.7 | 15.7 | 12.6 | 1.2 | 18.3 | 18.2 | 3.3 | 3.8 | 3.8 |
| BM25 | 9.5 | 12.5 | – | 13.6 | 12.8 | – | 13.4 | 7.4 | – | – | – | 9.4 | 8.8 | 2.8 | – | 3.3 |
| Closed | 4.7 | 5.6 | 5.8 | 5.3 | 5.5 | 4.0 | 4.4 | 5.5 | 5.9 | 5.3 | 1.9 | 4.1 | 3.8 | 2.6 | 2.3 | 2.4 |

"cn": "Zh-cn" (Chinese, simplified). "hk": "Zh-hk" (Chinese, Hong Kong). "tw":"Zh-tw" (Chinese, traditional).

Table 3: F1 scores on MKQA in mGEN-seen and unseen languages.

outperforms the baselines in all languages by a large margin except for Arabic and English. Note that Longpre et al. (2020) report results in a simplified setting with gold reference articles from the original Natural Questions dataset given in advance, and thus their results are not comparable. CORA yields larger improvements over the translate-test baseline in the languages that are distant from English and with limited training data such as Malay (Ms; 27.8 vs. 12.6) and Hebrew (He; 15.8 vs. 8.9). The performance drop of the translate-test model from English (43.3 F1) to other languages indicates the error propagation from the translation process. BM25 performs very poorly in some low-resource languages such as Thai because of the lack of answer content in the target languages' Wikipedia. MT+Mono underpeforms the MT+DPR baseline in MKQA since it is challenging to rerank answers from two separate methods with uncaliberated confidence scores. In contrast, CORA retrieves passages across languages, achieving around 30 F1 on a majority of the 26 languages.

## 4.2 Analysis

| Setting | XOR-TYDI QA | | | | MKQA | | | | | |
|---|---|---|---|---|---|---|---|---|---|---|
| | Avg. F1 | Ar | Ja | Te | Avg. F1 | Fi | Ru | Es | Th | Vi |
| CORA | **31.4** | **42.6** | **33.4** | **26.1** | **22.3** | **25.9** | **20.6** | **33.2** | 6.3 | **22.6** |
| (i) $\text{mDPR}_1$ + $\text{mGEN}_1$ | 27.9 | 36.2 | 29.8 | 21.1 | 17.3 | 23.1 | 13.1 | 28.5 | 5.7 | 18.6 |
| (ii) DPR (trained NQ)+mGEN | 24.3 | 30.7 | 29.2 | 19.0 | 17.9 | 20.1 | 16.9 | 29.4 | 5.5 | 18.2 |
| (iii) CORA, $\mathbf{C}^{multi}$={En} | 19.1 | 20.5 | 23.2 | 11.5 | 20.5 | 24.7 | 15.4 | 28.3 | **8.3** | 21.9 |
| (iv) mDPR+Ext.reader+MT | 11.2 | 11.8 | 10.8 | 5.6 | 12.2 | 16.1 | 10.9 | 25.2 | 1.2 | 12.7 |

Table 4: Ablation studies on XOR-TYDI QA development set and a subset of MKQA.

**Ablations: Impact of CORA components.** We compare CORA with the following four variants to study the impact of different components. (i) **$\text{mDPR}_1$ + $\text{mGEN}_1$** only trains CORA using the initial labeled, annotated data and measures the impact of the iterative training. (ii) **DPR (trained NQ) + mGEN** replaces mDPR with a multilingual BERT-based DPR trained on English data from Natural Questions (NQ), and encodes all passages in $\mathbf{C}^{multi}$. This configuration assesses the impact of cross-lingual training data. (iii) **CORA, $\mathbf{C}^{multi}$={En}** only retrieves from English during inference. This variant evaluates if English reference documents suffice to answer multilingual questions. (iv) **mDPR+Ext.reader+MT** replaces mGEN with an extractive reader model (Karpukhin et al., 2020) followed by answer translation. This variant quantifies the effectiveness of using a multilingual generation model over the approach that combines an extractive reader model with language-specific translation models. Note that for MKQA experiments, we sample the same 350 questions (~5%) from the evaluation set for each language to reduce the computational cost over varying configurations.

Figure 3: Breakdown of the languages of retrieved reference passages for sampled MKQA questions (%). The x and y axes indicate target (question) and retrieval reference languages respectively.

|       | es | fi | sv | ja | ko | ru | he | th | pt | no | zh | km | ms | tr |
|-------|------|------|------|------|------|------|------|------|------|------|------|------|------|------|
| en | 34.0 | 50.4 | 54.5 | 54.5 | 77.4 | 30.5 | 47.7 | 0.5 | 72.1 | 80.2 | 91.4 | 34.0 | 71.3 | 88.8 |
| es | 62.7 | 5.0 | 1.7 | 5.7 | 1.6 | 3.6 | 0.8 | 0.1 | 22.8 | 6.8 | 2.7 | 3.5 | 4.2 | 4.9 |
| fi | 0.5 | 40.8 | 1.7 | 0.2 | 0.2 | 0.3 | 0.3 | 0.0 | 0.6 | 1.6 | 0.3 | 1.4 | 0.8 | 1.1 |
| sv | 0.3 | 1.3 | 38.1 | 0.3 | 0.4 | 0.6 | 0.2 | 0.0 | 0.7 | 7.4 | 0.3 | 0.7 | 0.6 | 0.7 |
| ja | 0.0 | 0.1 | 0.0 | 37.8 | 0.0 | 0.1 | 0.1 | 0.0 | 0.0 | 0.0 | 2.9 | 0.3 | 0.0 | 0.1 |
| ko | 0.1 | 0.1 | 0.1 | 0.1 | 18.9 | 0.1 | 0.1 | 0.0 | 0.1 | 0.1 | 0.1 | 0.1 | 0.1 | 0.1 |
| ru | 0.9 | 0.9 | 1.1 | 0.6 | 0.3 | 64.0 | 0.5 | 0.0 | 0.9 | 1.0 | 0.5 | 0.4 | 0.5 | 0.7 |
| he | 0.4 | 0.2 | 0.5 | 0.2 | 0.1 | 0.1 | 48.8 | 0.0 | 0.6 | 0.3 | 0.2 | 0.2 | 0.3 | 0.2 |
| th | 0.0 | 0.0 | 0.0 | 0.0 | 0.2 | 0.0 | 0.0 | 99.4 | 0.0 | 0.0 | 0.0 | 58.1 | 0.0 | 0.0 |
| id | 0.8 | 1.1 | 2.1 | 0.6 | 0.7 | 0.6 | 1.1 | 0.0 | 1.7 | 2.3 | 1.6 | 1.2 | 22.2 | 3.1 |

|                    | Ja | Es |
|--------------------|----|----|
| retrieval errors   | 28 | 48 |
| different lang     | 18 | 0  |
| incorrect answer   | 22 | 36 |
| annotation error   | 22 | 12 |
| underspecified q   | 10 | 4  |

Table 6: Error categories (%) on 50 errors sampled from Japanese (Ja) and Spanish (Es) data.

Results in Table 4 show performance drops in all variants. This supports the following claims: (i) the iterative learning and data mining process is useful, (ii) mDPR trained with cross-lingual data substantially outperforms DPR with multilingual BERT trained on monolingual data only, (iii) reference languages other than English are important in answering multilingual questions, and (iv) a multilingual generation model substantially boosts the model performance.

| Setting | | **mDPR-Seen** | | | | | **Unseen** | | | | | |
|---------|------|------|------|------|------|------|------|------|------|------|------|------|
| Lang | Es | Fi | Ja | Ru | Th | Pt | Ms | Tr | Zh-Cn | Zh-Hk | Km |
| Script | Latn | | Jpan | Cyrl | Thai | Latn | | | Hant | | Khmr |
| mDPR $R^L$@10 | 53.7 | 52.8 | 32.9 | 42.3 | 14.9 | 50.0 | 49.4 | 42.0 | 12.6 | 16.6 | 15.7 |
| mDPR $R^{multi}$@10 | 63.4 | 60.9 | 42.0 | 54.0 | 28.0 | 62.6 | 63.4 | 55.4 | 40.6 | 42.3 | 25.1 |
| DPR(NQ) $R^L$@10 | 52.3 | 46.0 | 24.6 | 36.0 | 12.6 | 45.7 | 48.8 | 32.0 | 9.1 | 14.0 | 13.4 |
| DPR(NQ) $R^{multi}$@10 | 63.1 | 53.1 | 32.9 | 49.1 | 29.4 | 56.8 | 58.0 | 44.0 | 36.3 | 39.4 | 23.4 |

Table 5: Retrieval recall performance on MKQA as the percentage of the questions where at least one out of the top 10 passages includes an answer string in the target language ($R^L$@10), or in any language ($R^{multi}$@10). The same subset of the MKQA evaluation data are used as in the ablations.

**Retrieval performance and relationship to the final QA performance.** We evaluate CORA's retrieval performance on MKQA using two recall metrics that measure the percentage of questions with at least one passage among the top 10 that includes a string in an answer set in the target language ($R^L$@10) or in the union of answer sets from all languages that are available in MKQA ($R^{multi}$@10). MKQA provides answer translations across 26 languages.

Table 5 reports retrieval results for **mDPR** and multilingual BERT-based DPR trained on NQ: **DPR (NQ)**. This is equivalent to (ii) from the ablations. We observe that mDPR performs well in Indo-European languages with Latin script, even when the language is unseen. Interestingly, there is a significant performance gap between $R^L$@10 and $R^{multi}$@10 in languages with non-Latin script (e.g., Japanese, Russian, Chinese); this suggests that our model often uses relevant passages from other languages with Latin script such as English or Spanish to answer questions in those languages with non-Latin script. Our mDPR outperforms DPR (NQ) by a large margin in unseen languages with limited resources, which are consistent with the findings in Table 3. Nevertheless, we still see low performance on Khmer and Thai even with the $R^{multi}$@10 metric. We also observe that passage and query embeddings for those languages are far from other languages, which can be further studied in future work. We provide a two-dimensional visualization of the encoded passage representations in the appendix.

**Breakdown of reference languages.** Fig. 3 breaks down retrieved reference languages for each target language. Our multilingual retrieval model often retrieves documents from the target language (if its reference documents are available), English, or its typologically similar languages. For example, mDPR often retrieves Spanish passages for Portuguese questions and Japanese passages for Chinese questions; while they are considered phylogenetically distant, Japanese and Chinese overlap in script.

To further evaluate this, we conduct a controlled experiment: we remove Spanish, Swedish and Indonesian document embeddings and evaluate CORA on related languages: Danish, Portuguese and

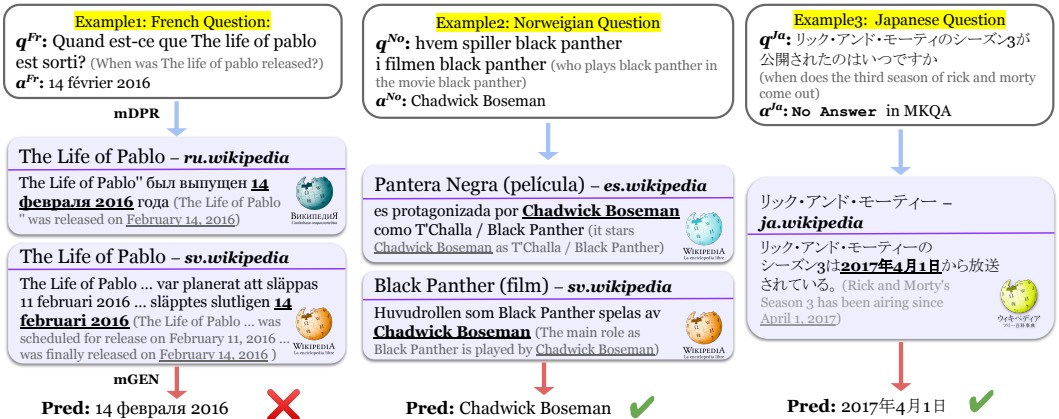

Figure 4: Cross-lingual retrieval and generation examples for three MKQA questions.

Malay. We observe performance drops of 1.0 in Danish, 0.6 in Portuguese, and 3.4 F1 points in Malay. This illustrates that while CORA allows for retrieval from any language in principle (*many-to-many*), cross-lingual retrieval from closer languages with more language resources is particularly helpful.

**Error analysis and qualitative examples.** Table 6 analyzes errors from CORA by manually inspecting 50 Japanese and Spanish wrong predictions from MKQA. We observe six major error categories: (a) retrieval errors, (b) generating correct answers in a different language (different lang), (c) incorrect answer generation (incorrect answer), (d) answer annotation errors (e.g., a correct alias isn't covered by gold answers, or Wikipedia information is inconsistent with English.), and (e) ambiguous or underspecified questions such as "who won X this year" (underspecified q). The table shows that both in Japanese and Spanish, the retrieval errors are dominant. In Japanese, CORA often generates correct answers in English, not in Japanese (different lang).

Fig. 4 shows some qualitative examples. The first example shows an error in (b): mGEN is generating an answer in Russian, not in French though the answer itself is correct. This type of error happens especially when retrieved passages are in languages other than the target and English.

**Human evaluation on cross-lingual retrieval results.** To observe how cross-lingual retrieval between distant languages is actually helping, we sample 25 Norwegian questions for which Spanish passages are included among the top 10 retrieved results. As seen in Fig. 3, CORA retrieves Spanish (es) passages for 6.8% of the Norwegian (no) questions. A Spanish speaker judges if the retrieved Spanish passages actually answer the given Norwegian questions.[11] We found that in 96% of the cases, the retrieved Spanish passages are relevant in answering the question. One such example is presented in Fig. 4 (the second example).

**Human analysis on unanswerable questions.** CORA retrieves passages from a larger multilingual document collection than the original human annotations. Thus, CORA may further improve the answer coverage over the original human annotations. MKQA includes questions that are marked as unanswerable by native English speakers given English knowledge sources. We sample 400 unanswerable Japanese questions whose top one retrieved passage is from a non-English Wikipedia article. Among these, 329 unanswerable questions are underspecified (also discussed in Asai and Choi, 2021). For 17 out of the 71 remaining questions, the answers predicted by CORA are correct. This finding indicates the significance of cross-lingual retrieval and potential room for improvement in annotating multilingual open QA datasets. The third example in Fig. 4 shows one of these cases.

## 5   Related Work and Broader Impacts

**English and non-English open QA.** Despite the rapid progress in open QA (Chen et al., 2017; Karpukhin et al., 2020), most prior work has been exclusively on English (Lewis et al., 2020; Izacard and Grave, 2021b). Several prior attempts to build multilingual open QA systems often rely on machine translation or language-specific retrieval models (Ture and Boschee, 2016; Asai et al., 2021).

---

[11] During evaluation, we provide the original English questions from MKQA.

Lewis et al. (2020) and Guu et al. (2020) introduce *retrieve-then-generate* approaches. Izacard and Grave (2021a) introduce an iterative training framework that uses attention weights from a generator model as a proxy for text relevance scores. Tran et al. (2020) introduce CRISS, a self-supervised pre-training approach consisting of a parallel sentence mining module and a sequence-to-sequence model, which are trained iteratively. Several recent work such as Xiong et al. (2021) improves DPR by mining and learning with hard examples. Our work is the first work that introduces a unified multilingual system for *many-to-many* open QA, which is a challenging task requiring massive-scale cross-lingual retrieval and has not been addressed in prior work. We introduce an iterative training and data mining approach guided by filtering from an answer generation model to automatically extend annotated data available only in high-resource languages to low-resource. This approach contributes to significant performance improvements in languages without annotated training data.

**Many-languages-one models.** Several recent work introduces single multilingual models for many languages using pre-trained multilingual models such as mBERT or mT5 in many NLP tasks (e.g., entity linking: Botha et al., 2020; De Cao et al., 2021; semantic role labeling: Mulcaire et al., 2019b; Lyu et al., 2019; Fei et al., 2020; syntactic parsing: Mulcaire et al., 2019a; Kondratyuk and Straka, 2019). This work conducts the first large-scale study of a unified multilingual open QA model across many languages and achieves state-of-the-art performance in 26 typologically diverse languages.

**Synthetic data creation for machine reading comprehension.** Alberti et al. (2019) introduce a method of generating synthetic machine reading comprehension data by automatically generating questions and filtering them out by a trained machine reading comprehension model. Several studies augment multilingual machine reading comprehension training data by generating new question-answer pairs from randomly sampled non-English Wikipedia paragraphs (Riabi et al., 2021; Shakeri et al., 2020). This work focuses on multilingual open QA, which involves not only machine reading comprehension but also cross-lingual retrieval. A similar augmentation method for machine reading comprehension can be applied to further improve the answer generation component in CORA.

**Societal impacts.** Our code and data are publicly available. CORA can perform open QA in unseen languages and can benefit society in building QA systems for low-resource languages, hence enabling research in that direction. Unlike previous models, CORA removes the necessity of external black-box APIs, and thus we can examine and address wrong answers due to model errors or misinformation present on Wikipedia. This would help us mitigate the potential negative impact from CORA or its subsequent models outputting a wrong answer when it is used by people who seek information.

## 6   Conclusion

To address the information needs of many non-English speakers, a QA system has to conduct cross-lingual passage retrieval and answer generation. This work presents CORA, a unified multilingual *many-to-many* open QA model that retrieves multilingual passages in many different languages and generates answers in target languages. CORA does not require language-specific translation or retrieval components and can even answer questions in unseen, new languages. We conduct extensive experiments on two multilingual open QA datasets across 28 languages, 26 of which CORA advances the state of the art on, outperforming competitive models by up to 23 F1 points. Our extensive analysis and manual evaluation reveal that CORA effectively retrieves semantically relevant passages beyond language boundaries, and can even find answers to the questions that were previously considered unanswerable due to lack of sufficient evidence in annotation languages (e.g., English). Nonetheless, our experimental results show that the retrieval component still struggles to find relevant passages for queries in some unseen languages. Our analysis also showed that CORA sometimes fails to generate an answer in the target language. In future work, we aim to address these issues to further improve the performance and scale our framework to even more languages.

## Acknowledgement

This research was supported by NSF IIS-2044660, ONR N00014-18-1-2826, gifts from Google, the Allen Distinguished Investigator Award, the Sloan Fellowship, and the Nakajima Foundation Fellowship. We thank anonymous reviewers, area chairs, Eunsol Choi, Sewon Min, David Wadden, and the members of the UW NLP group for their insightful feedback on this paper, and Gabriel Ilharco for his help on human analysis.

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
