# Appendix

## A  Details of Modeling

### A.1  Input format

**Passage representations.**    To create a passage representation, the passage title and text are concatenated (`[CLS]` *title* `[SEP]` *passage* `[SEP]`), following common practice (Karpukhin et al., 2020). We retrieve top 10 passages and use them as input to mGEN.

**Generator input.**    The input to the generator is a concatenation of $q^L$ and $\mathcal{P}^{multi}$. As described in § 2.1, we append a language tag that represents $L$ to $q^L$. For each passage, we prepend the retrieved ranks and the original Wikipedia article titles and concatenate them to form a input paragraph sequence. We differentiate those paragraphs from the question using special tokens (`<P>` vs. `<Q>`). Finally, the concatenated passages are appended to $q^L$ and the language tag. Below is an example input:

*<Q>:* ロンポールの学部時代の専攻は何？*[ja] <P>:<0:*ロン・ポール*>*ロナルド・アーネスト・ポール *(英語: Ronald Ernest "Ron" Paul、1935年8月20日 - )* は、アメリカ合衆国の元政治家。共和党所属でテキサス州選出の元連邦下院議員であった *<1: Ron Paul> Paul went to Gettysburg College, where he was a member of the Lambda Chi Alpha fraternity. He graduated with a B.S. degree in* **Biology** *in 1957.*

As in the case of machine translation, we found that the language code does not need to be specified during inference as our model learns the question language automatically. Yet, we found that training with language codes is particularly useful to augment training data for $L_{target}$ without any question data in $L_{target}$. In particular, given questions from existing datasets in $L_{source}$ and entities names in $L_{target}$ corresponding to the original answers in $L_{source}$, our generator learns to generate answers in $L_{target}$ from the language code even when questions themselves are written in $L_{source}$. Please see the details of training mGEN with synthetic data in the next section.

### A.2  Details of the Data Mining Process

**Synthetic data for mGEN.**    To train mGEN to generate answers in languages that are not covered by annotated data or our reference sources, we augment English QA data ($q^{En}, a^{En}$) from Natural Questions (Kwiatkowski et al., 2019). We first use an English DPR model to retrieve $\mathcal{P}^{En}$ for each $q^{En}$. Then, we automatically translate $a^{En}$ to a target language $L$ using Wikipedia language links. We use Media Wiki API,[12] and form new mGEN training data ($q^{En}, a^L, \mathcal{P}^{En}$). Although the questions and passages are all written in English, our model knows in which language it should answer from the language code appended to the question. We limit the target languages for this augmentation process to Arabic, Finnish, Japanese, Korean, Russian, Spanish, Swedish, Hebrew, Thai, Danish, French, Italian, Dutch, Polish, and Portuguese. Interestingly, just adding this language code effectively changes the outputs as shown in Table 7. Although we could create at most 15 synthetic data for each($q^{En}, a^{En}, \mathcal{P}^{En}$), we sample at most 10 languages from the 15 languages to avoid overfitting. We further subsample 50% of the synthetically generated questions. Those synthetically generate data is introduced after training mGEN for 3 epochs to avoid overfitting.

| input question | output | gold answers |
|---|---|---|
| who is the actor that plays the good doctor [ja] | フレッド・ハイモア | フレディ・ハイモア |
| who is the actor that plays the good doctor [ko] | 프레디 하이모어 | 프레디 하이모어 |
| who is the actor that plays the good doctor [it] | Freddie Highmore | Freddie Highmore |

Table 7: Examples of mGEN outputs with varying language codes.

---

[12]`https://www.wikidata.org/w/api.php`.

| language | The number of articles | The number of passages |
|---|---|---|
| English | 6,297,085 | 18,003,200 |
| Arabic | 664,693 | 1,304,828 |
| Finnish | 451,338 | 886,595 |
| Japanese | 1,268,148 | 5,116,905 |
| Korean | 441,316 | 638,864 |
| Russian | 1,522,499 | 4,545,635 |
| Bengali | 64,556 | 179,936 |
| Telugu | 70,356 | 274,230 |
| Indonesian | 452,304 | 820,572 |
| Thai | 129,122 | 520,139 |
| Hebrew | 237,836 | 1,045,255 |
| Swedish | 3,758,071 | 4,525,695 |
| Spanish | 1,453,732 | 5,738,484 |

Table 8: Statistics of the Wikipedia data.

## B    Details of Experiments

### B.1    Details of the knowledge source language selection.

In addition to the English Wikipedia embeddings, we encode all of the passages from the Wikipedias of all of the ten languages included in XOR-TYDI QA or TYDI QA. Adding all of the languages available in Wikipedia to our document collection would significantly increase the index size and slow down inference. Therefore, we add the languages among the 26 MKQA languages that satisfy the following criteria: (i) a language is included in XOR-TYDI QA or TYDI QA, (ii) a language uses non-Latin script and has the largest number of the Wikipedia articles among the languages in the same language family branch (e.g., Thai), or (iii) a language uses Latin script and has more than 1.5 million articles as of May 2021.[13]

### B.2    Details of Wikipedia statistics

For our multilingual retriever, we split each article into 100-token chunks (Karpukhin et al., 2020), while BM25 first splits Wikipedia articles into the pre-defined paragraph units. We also filter out the short articles with fewer than $k$ (i.e., $k = 20$ in this work) tokens, following common techniques in open QA (Min et al., 2021) in $\mathbf{C}^{multi}$. As a result, we add more than 43.6 million articles across the languages. The original passage text file is 29GB, and the total index size is around 129 GB.

### B.3    Licence, ethical considerations and data splits of XOR-TYDI QA and MKQA

**Licence.**    Both two datasets are under the MIT licence. The dataset can be downloaded from their official repositories.[14]

**Potential risk of offensive or personally identifiable information.**    MKQA (Longpre et al., 2020) questions are originally from the Natural Questions data (Kwiatkowski et al., 2019). The questions are anonymized Google Search queries, and we expect that those questions are not personally identifiable. The Natural Questions authors conduct several procedures to filter out noisy questions, and we expect that the questions do not contain offensive or inappropriate content.

Likewise, XOR-TYDI QA questions are from TYDI QA, where questions are written by their in-house annotates who have native proficiency in the target languages. The TYDI QA authors trained those in-house annotators and asked them to write questions that they are interested in given short prompts. We expect that the resulting questions are not personally identifiable and have no risk of offensive information.

---

[13] https://en.wikipedia.org/wiki/List_of_Wikipedias.

[14] https://github.com/apple/ml-mkqa for MKQA; https://github.com/AkariAsai/XORQA for XOR-TYDI QA.

|  | **Ar** | **Bn** | **Da** | **De** | **En** |
|---|---|---|---|---|---|
| name | Arabic | Bengali | Danish | German | English |
| family | Afro-Asiatic | Indo-European | Indo-European | Indo-European | Indo-European |
| branch | Semitic | Indo-Iranian | Germanic | Germanic | Germanic |
| script | Arab | Beng | Latn | Latn | Latn |
|  | **Es** | **Fi** | **Fr** | **He** | **Hu** |
| name | Spanish | Finnish | French | Hebrew | Hungarian |
| family | Indo-European | Uralic | Indo-European | Afro-Asiatic | Uralic |
| branch | Italic | Finic | Italic | Semitic | Finno-Ugric |
| script | Latn | Latn | Latn | Hebr | Latn |
|  | **It** | **Ja** | **Ko** | **Km** | **Ms** |
| name | Italian | Japanese | Korean | Khmer | Malay |
| family | Indo-European | Japonic | Koreanic | Austroasiatic | Austronesian |
| branch | Italic | Japanese | Korean | Proto-Mon-Khmer | Malayo-Polynesian |
| script | Latn | Jpan | Hang | Khmr | Latn |
|  | **Nl** | **No** | **Pl** | **Pt** | **Ru** |
| name | Dutch | Norwegian | Polish | Portuguese | Russian |
| family | Indo-European | Indo-European | Indo-European | Indo-European | Indo-European |
| branch | Germanic | Germanic | Balto-Slavic | Italic | Balto-Slavic |
| script | Latn | Latn | Latn | Latn | Cyrl |
|  | **Sv** | **Te** | **Th** | **Tr** | **Vi** |
| name | Swedish | Telugu | Thai | Turkish | Vietnamese |
| family | Indo-European | Dravidian | Kra–Dai | Altaic | Austroasiatic |
| branch | Germanic | South-Centra | Tai | Turkic | Vietic |
| script | Latn | Telu | Thai | Latn | Latn |
|  | **Zh-cn** | **Zh-hk** | **Zh-tw** |  |  |
| name | Chinese (Simplified) | Chinese (Hong Kong) | Chinese (Traditional) |  |  |
| family | Sino-Tibetan | Sino-Tibetan | Sino-Tibetan |  |  |
| branch | Chinese | Chinese | Chinese |  |  |
| script | Hans/Hant | Hant | Hant |  |  |

Table 9: List of 28 language we test in this work. The script is based on ISO 15924.

**Data splits.** MKQA does not have any train data, and we use the questions with answer annotations for evaluation, removing 1,427 `unanswerable` questions and 1,815 `long_answer` questions. Consequently, MKQA evaluation data has 6,758 questions for each target language. Note that the `unanswerable` and `long_answer` type information is provided in the original MKQA dataset, and we do not conduct any manual data filtering. For the ablation or controlled experiments, we randomly sample 350 questions from the 6,758 questions with short answers due to our computational constraints.

We use the train, dev and test data splits from the original XOR-TYDI QA (full) data.

### B.4 Language family, branch and script type Information of the languages

Table 9 provides a full list of the 28 languages and their language family, branch and script type information. The target languages are typologically diverse; 12 of them use their own script system, which makes answer generation in those languages harder than in the languages with Latin script.

### B.5 Hyperparameters of CORA

**mDPR.** We first fine-tune mDPR on the Natural Questions data using the training data file released by DPR authors.[15] We filter out questions that are used to create MKQA evaluation data by comparing the input questions. We use the same hyperparameters as in the original DPR (Karpukhin et al., 2020). We then perform fine-tuning on TYDI QA and XOR-TYDI QA's gold paragraph data, initializing the checkpoint that achieves the best performance on the development data. We fine-tune the model for 40 epochs and use the checkpoint that produces the best retrieval performance on XOR-TYDI QA's development data. We use 8 GPUs with 24G RAM, and the total batch size is 128. We empirically

---

[15] https://github.com/facebookresearch/DPR.

| hyperparameter | |
| --- | --- |
| max source length | 1,000 |
| max target length | 25 |
| batch size (per GPU) | 2 |
| label smoothing | 0.1 |
| dropout | 0.1 |
| warmup steps | 500 |
| learning rate | 3e-5 |
| weight decay | 0.001 |
| adam $epsilon$ | 1e-8 |
| max grad norm | 0.1 |
| gradient accumulation steps | 2 |

Table 10: Hyperparameters of mGEN.

found that using the updated query encoder hurt the retrieval performance on MKQA, while in XOR-TYDI QA we observe a marginal performance drop. Therefore, at inference, we continue using the query encoder trained on the initial data, while we use the updated passage encoder to encode $\mathbf{C}^{multi}$.

**mGEN.** The full list of the hyperparameters are in Table 10. We first train our mGEN using the initial data for 15 epochs and use the checkpoint that gives the highest development score. We use Adam (Kingma and Ba, 2015) for optimization. We subsequently apply iterative training. During our $t$th iterative step, we use the best checkpoint so far to label new positive and negative passages, which will then be used to fine-tune mDPR at the next iteration. After we finish mGEN training at the $t$-th iteration, we use the best checkpoint for the final evaluation without performing additional data mining. We use our internal cluster to run all of the mGEN related training. We use 8 GPUs with 24G RAM, and the total batch size is 32.

**Inference.** During inference, we first retrieve top 15 passages using mDPR, and then feed the questions and concatenated passages into the mGEN model, with language tags. We use the same checkpoints and encoded embeddings for MKQA and XOR-TYDI QA. There are minor differences in the gold answer format in MKQA and XOR-TYDI QA due to different annotation methods (e.g., translate English answers by Wikidata vs. use answers extracted from the target language Wikipedias). One may fine-tune different models using different subsets of training examples (e.g., MKQA can benefit from more NQ-based synthetic training data as the questions are originally from NQ). In this work, we focus on building a unified QA system across languages and datasets, and thus use the same checkpoints for all of the experiments.

### B.6 Details of translate-test baseline

We first translate the MKQA and XOR-TYDI QA questions from various languages to English, use DPR to retrieve the answers from the knowledge source, use the reader to extract an answer, and then translate the answer back to its original language.

**Details of translation models.** Our translation models (to English and from English) are the pretrained MarianMT (Junczys-Dowmunt et al., 2018) style OPUS-MT (Tiedemann and Thottingal, 2020) models available in Transformers library (Wolf et al., 2020) that are trained on the OPUS corpus (Tiedemann and Nygaard, 2004). Since there is no MarianMT pre-trained OPUS-MT model from English to Korean on Transformers, we use the pre-trained base-sized autoregressive transformers model provided by the authors of XOR-TYDI QA.[16]

Some of the newer OPUS-MT models require a prefix of the target language before each sentence of the source language (English here) when translating English answers back to the question's original language, which is usually the ISO 639-3 language code. For example, if we want to translate *Ron Paul* from English to Arabic, we concatenate the prefix "»ara«" and the original sentence together

---

[16]`https://github.com/jungokasai/XOR_QA_MTPipeline`.

|  | **Ar** | **Da** | **De** | **Es** |
|---|---|---|---|---|
| To-English Model | `opus-mt-ar-en` | `opus-mt-da-en` | `opus-mt-de-en` | `opus-mt-es-en` |
| From-English Prefix | »ara« | N/A | N/A | N/A |
| From-English Model | `opus-mt-en-ar` | `opus-mt-en-da` | `opus-mt-en-de` | `opus-mt-en-es` |
|  | **Fi** | **Fr** | **He** | **Hu** |
| To-English Model | `opus-mt-fi-en` | `opus-mt-fr-en` | `opus-mt-afa-en` | `opus-mt-hu-en` |
| From-English Prefix | N/A | N/A | »heb« | N/A |
| From-English Model | `opus-mt-en-fi` | `opus-mt-en-fr` | `opus-mt-en-afa` | `opus-mt-en-hu` |
|  | **It** | **Ja** | **Ko** | **Km** |
| To-English Model | `opus-mt-it-en` | `opus-mt-ja-en` | `opus-mt-ko-en` | `opus-mt-mul-en` |
| From-English Prefix | N/A | N/A | N/A | »khm_Latn« |
| From-English Model | `opus-mt-en-it` | `opus-mt-en-jap` | N/A | `opus-mt-en-mul` |
|  | **Ms** | **Nl** | **No** | **Pl** |
| To-English Model | `opus-mt-mul-en` | `opus-mt-nl-en` | `opus-mt-gem-en` | `opus-mt-pl-en` |
| From-English Prefix | »zsm_Latn« | N/A | »nno« | »pol« |
| From-English Model | `opus-mt-en-mul` | `opus-mt-en-nl` | `opus-mt-en-gem` | `opus-mt-en-sla` |
|  | **Pt** | **Sv** | **Th** | **Tr** |
| To-English Model | `opus-mt-ROMANCE-en` | `opus-mt-sv-en` | `opus-mt-th-en` | `opus-mt-tr-en` |
| From-English Prefix | »pt« | N/A | »tha« | »tur« |
| From-English Model | `opus-mt-en-ROMANCE` | `opus-mt-en-sv` | `opus-mt-en-mul` | `opus-mt-en-trk` |
|  | **Vi** | **Zh-cn** | **Zh-hk** | **Zh-tw** |
| To-English Model | `opus-mt-vi-en` | `opus-mt-zh-en` | `opus-mt-zh-en` | `opus-mt-zh-en` |
| From-English Prefix | »vie« | »cmn« | »yue_Hant« | »cmn_Hant« |
| From-English Model | `opus-mt-en-vi` | `opus-mt-en-zh` | `opus-mt-en-zh` | `opus-mt-en-zh` |

Table 11: Translation models and prefixes used for the translate-test baseline.

to specify the target language to be Arabic since the `opus-mt-en-ar` [17] model supports multiple target languages. Then, we feed the concatenated result *"»ara« Ron Paul"* into the translation model and get the translation.

Such prefixes and the models we use for each language are listed in Table 11.

**Details of English DPR model.** For the English DPR model, we use the trained retriever and reader models from XOR-TYDI QA, which can be downloaded from their official website.[18]

## B.7 Details of BM25 baseline

We use the February 2019 Wikipedia dumps as the knowledge source and retrieval corpus for our BM25 baseline. We first use wikiextractor to preprocess the Wikipedia documents, and then use Pyserini (Lin et al., 2021), which relies on Apache Lucene 8.0.0[19] to index the documents and retrieve the BM25 results for XOR-TYDI QA and MKQA questions. We use 2 paragraphs as one basic unit of retrieval where paragraphs are separated by '\n'. We retrieve the top 10 units that have the highest BM25 score. After we retrieve top units, we concatenate those paragraphs and feed them into a bert-base-multilingual-uncased extractive QA model that predicts start and end positions. The final answers are determined as the span with the highest joint probabilities.

French, Hebrew, Khmer, Malay, Polish, Vietnamese and Chinese (Hong Kong) are either not supported by Apache Lucene or missing from the Wikipedia dumps, and therefore are not included in the final results.

## B.8 Details of MT+Mono baseline

We normalize the predicted probabilities from the the BM25 (monolingual) baseline so that the score will be between 0 to 1. We output the monolingual baseline's answer when the probability is higher than a threshold; otherwise, we output translated answers from the translate-test baseline. In this work, we set the threshold to 0.5 given the results on the XOR-TYDI QA development set.

---

[17]`https://huggingface.co/Helsinki-NLP/opus-mt-en-ar`.

[18]`https://github.com/AkariAsai/XORQA/tree/main/baselines`.

[19]`https://lucene.apache.org/core/8_0_0/index.htmlhttps://lucene.apache.org/core/8_0_0/index.html`.

| Models | Target Language $L_i$ F1 | | | | | | | Macro Average | | |
|---|---|---|---|---|---|---|---|---|---|---|
| | **Ar** | **Bn** | **Fi** | **Ja** | **Ko** | **Ru** | **Te** | **F1** | **EM** | **BLEU** |
| CORA | **42.9** | 26.9 | **41.4** | **36.8** | **30.4** | **33.8** | **30.9** | **34.7** | **25.8** | **23.3** |
| GMT+GS | 18.0 | **29.1** | 13.8 | 5.7 | 15.2 | 14.9 | 15.6 | 16.0 | 9.9 | 14.9 |
| MT+ Mono | 15.8 | 9.6 | 20.5 | 12.2 | 11.4 | 16.0 | 0.5 | 17.3 | 7.5 | 10.7 |
| MT+DPR | 7.2 | 4.3 | 17.0 | 7.9 | 7.1 | 13.6 | 0.5 | 8.2 | 3.8 | 6.8 |
| BM25 | 18.4 | 14.9 | 18.8 | 12.7 | 12.1 | 13.5 | – | – | – | – |
| Closed-book | 14.0 | 8.1 | 11.8 | 19.1 | 9.3 | 10.5 | 7.6 | 11.5 | 8.2 | 4.9 |

Table 12: Performance on XOR-FULL (development data F1 scores and macro-averaged F1, EM, and BLEU scores). "GMT+GS" denotes the previous state-of-the-art model, which combines Google Custom Search in the target language and Google Translate + English DPR for cross-lingual retrieval (Asai et al., 2021). Pyserini does not support Telugu.

## B.9 Details of the Closed-book Baseline

Instead of training a sequence-to-sequence model from scratch using question and answer only data as in Roberts et al. (2020), we use the same mGEN model as in CORA, and only at inference time do we skip retrieval. We also tested an mt5-base based sequence-to-sequence model that is trained to generate answers from questions only, but this model underperformed the inference-only model on the XOR-TYDI QA development set.

## C Additional Results

### C.1 Results on XOR-TYDI QA Development Set

Table 12 shows the results on the XOR-TYDI QA development set. We clearly outperform the previous state-of-the-art model, as well as the competitive baselines, by a large margin across target languages. The scores on XOR-FULL development set are significantly higher than the XOR-FULL test set presented in Table 1. We have found that the proportions of the questions where answers can be extracted from the target languages' Wikipedia are significantly higher than in XOR-FULL test set. Our CORA framework improves the performance on those "in-language" subsets of XOR-FULL and get a large performance jump on the test set.

### C.2 EM Scores on XOR-TYDI QA and MKQA

**EM scores on XOR-TYDI QA.** The EM scores on the XOR-TYDI QA test data are in Table 13. We significantly outperform all other baselines and previous state-of-the-art models in all languages except for Korean. We found that in Korean, our model is often penalized because outputs are correct yet generated in English, not Korean. The state-of-the-art model ensures that the answers are in Korean using Google Translate, which helps the system to get high performance in Korean.

**EM scores on MKQA.** The EM scores on MKQA test set are shown in Tables 14 and 15. CORA outperforms the other baselines by large margins in all of the languages except for Arabic and English. Note that EM scores may underestimate the models' ability of open retrieval; generated answers may be correct even if they do not have a matching sub-span in existing documents, existing Wikidata entries, or human translated answers (Asai et al., 2021).

## D Further Analysis

### D.1 Visualizing the Document Embedding Spaces

We plot two dimensional encoded document representations (PCA) for the corresponding articles in Fig. 5. The gray dots concentrated in the lower right part in the first figure represent encoded Thai embeddings. As we can see from the plot before cross-lingual training, the Thai document embeddings are far apart from other languages' embeddings. On the other hand, after iterative

| Models | Target Language $L_i$ EM | | | | | | |
|---|---|---|---|---|---|---|---|
| | **Ar** | **Bn** | **Fi** | **Ja** | **Ko** | **Ru** | **Te** |
| CORA | **38.4** | **26.6** | **33.1** | **30.1** | 18.9 | **36.3** | **34.6** |
| SER | 23.0 | 16.1 | 18.5 | 6.3 | 9.3 | 6.9 | 14.4 |
| GMT+GS | 22.1 | 10.9 | 13.3 | 3.0 | **20.1** | 11.4 | 9.1 |
| MT+ Mono | 19.1 | 9.0 | 16.7 | 7.1 | 8.3 | 13.1 | 0.5 |
| MT+DPR | 2.5 | 1.5 | 10.4 | 3.3 | 2.9 | 2.5 | 0.5 |
| BM25 | 23.2 | 14.6 | 17.0 | 5.3 | 9.4 | 13.4 | – |
| Closed-book | 9.6 | 6.7 | 8.8 | 16.8 | 7.8 | 15.5 | 1.9 |

Table 13: Performance on XOR-FULL (test data EM scores). "GMT+GS" denotes the previous state-of-the-art model, which combines Google Custom Search in the target language and Google Translate + English DPR for cross-lingual retrieval (Asai et al., 2021). Concurrent to our work, "SER" is the current state-of-the-art model, Single Encoder Retriever, submitted anonymously on July 14 to the XOR-FULL leaderboard (https://nlp.cs.washington.edu/xorqa/). Pyserini does not support Telugu.

| Setting | – | Included in XOR-TYDI QA | | | | | | Seen by mDPR and mGEN | | | |
|---|---|---|---|---|---|---|---|---|---|---|---|
| | Avg. over all $L$. | **En** | **Ar** | **Fi** | **Ja** | **Ko** | **Ru** | **Es** | **Sv** | **He** | **Th** |
| CORA | **17.2** | 31.2 | 7.7 | **21.8** | **7.4** | **7.3** | **13.8** | **25.8** | **26.0** | **10.7** | 4.8 |
| MT+Mono | 9.6 | **32.1** | 3.9 | 12.6 | 3.8 | 4.0 | 6.5 | 15.2 | 14.4 | 4.4 | 5.0 |
| MT+DPR | 10.2 | 32.1 | **8.5** | 15.0 | 3.6 | 2.7 | 8.6 | 18.6 | 4.4 | 2.8 | 4.6 |
| BM25 | – | 12.8 | 3.1 | 6.5 | 2.7 | 3.9 | 4.7 | 9.4 | 6.6 | – | 2.6 |
| Closed | 2.3 | 3.4 | 2.0 | 2.0 | 3.1 | 2.0 | 2.3 | 3.1 | 4.4 | 2.0 | 2.6 |

Table 14: EM scores on MKQA seen languages over 6.7k questions with short answer annotations.

| Setting | Seen in mGEN | | | | | | | Unseen | | | | | | | | |
|---|---|---|---|---|---|---|---|---|---|---|---|---|---|---|---|---|
| | **Da** | **De** | **Fr** | **It** | **Nl** | **Pl** | **Pt** | **Hu** | **Vi** | **Ms** | **Km** | **No** | **Tr** | **cn** | **hk** | **tw** |
| CORA | **25.8** | **25.4** | **25.4** | **24.0** | **26.5** | **21.4** | **23.0** | **15.4** | **17.7** | **23.2** | 4.8 | **24.3** | **18.5** | **4.1** | **5.5** | **4.5** |
| MT+Mono | 13.2 | 15.8 | 15.4 | 15.3 | 15.3 | 14.6 | 13.9 | 11.5 | 7.4 | 3.9 | 0.1 | 11.6 | 11.2 | 3.5 | 2.8 | 4.1 |
| MT+DPR | 16.9 | 16.9 | 15.4 | 16.1 | 17.5 | 14.6 | 14.6 | 10.3 | 7.4 | 3.9 | 0.1 | 13.6 | 8.7 | 1.1 | 2.8 | 3.6 |
| BM25 | 5.8 | 8.7 | – | 9.4 | 8.4 | – | 8.7 | 4.8 | – | – | – | 5.6 | 5.6 | 2.5 | – | 2.8 |
| Closed | 2.6 | 3.2 | 2.6 | 2.6 | 2.8 | 2.4 | 2.6 | 2.3 | 2.2 | 2.3 | 1.4 | 2.3 | 1.7 | 2.1 | 1.8 | 1.9 |

"cn": "zh-cn" (Chinse, simplified). "hk": "zh-hk" (Chinese, Hong Kong). "tw": "zh-tw" (Chinese, traditional).

Table 15: EM scores on MKQA in languages unseen by mDPR and not included in $\mathbf{C}^{multi}$.

training (Fig. 5(b)), the embeddings from many languages get closer, though we can still see loose clusters of some languages.

## D.2 Analysis on the Cross-lingual Retrieval

**Spanish paragraphs retrieved for Norwegian questions.** The Spanish paragraphs retrieved for Norwegian questions are shown in Table 16. As we can see, CORA retrieves the Spanish passages relevant to the given Norwegian questions and generate correct answers. Although those two languages both belong to the Indo-European family and use Latin script, their typological properties (e.g., syntax and vocabulary) differ significantly.

## D.3 Analysis on Errors on MKQA Data.

**Details of the annotation process.** We randomly sample 50 errors from Spanish and Japanese, and we classify those errors into five categories described in § 4.2. Each sample includes: a question in the target language, a question in English (the original NQ question), the top one passage retrieved by mDPR, an answer generated by mGEN, and gold answers. The error analysis is conducted by bilingual or native speakers of Spanish or Japanese.

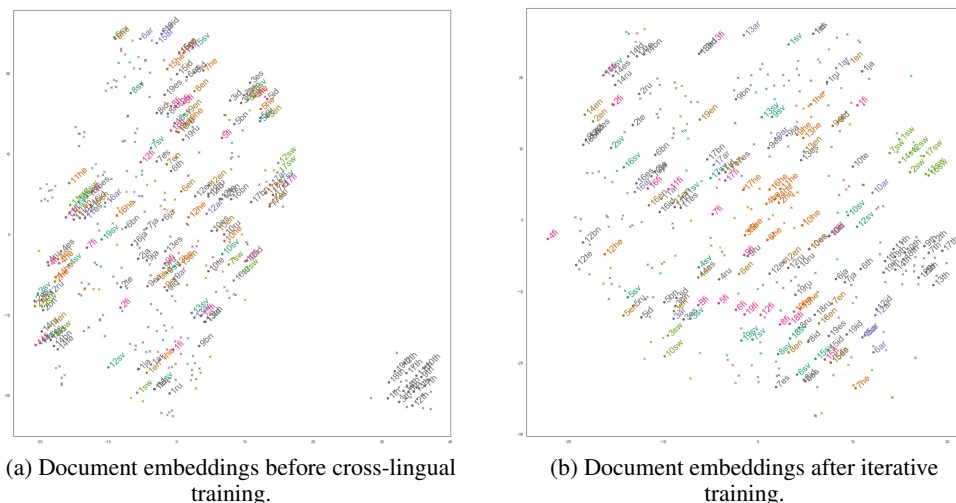

| (a) Document embeddings before cross-lingual training. | (b) Document embeddings after iterative training. |

Figure 5: Embeddings before and after cross-lingual training (PCA).

| Query | Paragraph | Gold Answer |
|---|---|---|
| hvem spilte maria magdalena i jesus christ superstar (trans: who played mary magdalene in jesus christ superstar) | Los actores principales de la película eran Ted Neeley en el papel de Jesús, Carl Anderson en el de Judas e **Yvonne Elliman** en el papel de María Magdalena. (**trans**: The main actors in the film were Ted Neeley as Jesus, Carl Anderson as Judas, and Yvonne Elliman as Mary Magdalene. ) | Yvonne Elliman |
| hvor mange episoder er det i andre sesong av my hero academia (trans: how many episodes are in season two of my hero academia) | El estreno fue 7 de abril del 2018. Contando con un total de 25 episodios igual que la segunda. (**trans**: The premiere was April 7, 2018. With a total of 25 episodes the same as the second.) | 25.0 episodes |

Table 16: Cross-lingual retrieval examples between Norweigian questions and Spanish passages that lead to correct answers.

|  | Japanese | Spanish | Chinese (simplified) |
|---|---|---|---|
| retrieval error | 28 | 48 | 70 |
| different lang | 18 | 0 | 10 |
| incorrect answer | 22 | 36 | 4 |
| annotation error | 22 | 12 | 4 |
| underspecified question | 10 | 4 | 12 |

Table 17: Error categories (%) on 50 errors sampled from Japanese, Spanish and Chinese (simplified) data. Error questions are sampled from the MKQA evaluation data.

**Error analysis results on Chinese examples.** We also conduct the same analysis in Chinese to understand the relatively low performance in the three Chinese languages. Among the three Chinese variants, we choose simplified Chinese (Zh-cn). The error analysis is conducted by a native speaker. Table 17 shows the error analysis results in Japanese, Spanish, and Chinese. Generating answers in different languages is common in Chinese like Japanese. Retrieval errors account for 70% of the errors in Chinese, which is significantly higher than the proportions in Japanese or Spanish. This can be explained by the fact that we do not include Chinese passages in our $\mathbf{C}^{multi}$, and thus CORA always has to conduct cross-lingual retrieval to get evidence to answer. Retrieving documents cross-lingually is more challenging than retrieving documents monolingually, and future work can improve cross-lingual retrieval particularly between languages that are distant from each other.

| error type | Query | Paragraph | Prediction [gold Answer] |
|---|---|---|---|
| different lang | マルコム in the Middleで父親役は誰でしたか (**trans**: who played the dad on malcolm in the middle?) | The series is about a boy named Malcolm (Frankie Muniz), the third-born child in a comically dysfunctional working-class family of four, and later, five boys, the sons of Lois (Jane Kaczmarek) and Hal (Bryan Cranston). | Bryan Cranston [ブライアン・クランストン (**trans**:Bryan Cranston)] |
| incorrect answer | 「愛はとまらない」を唄っているのは誰ですか (**trans**: who sings nothing's gonna stop us now?) | 愛はとまらないはアルバート・ハモンドとダイアン・ウォーレンの共作による楽曲。アメリカ合衆国のロックバンド、スターシップにより録音された。 (**trans**:Nothing's Gonna Stop is a song co-written by Albert Hammond and Diane Warren. Recorded by the American rock band Starship.) | アルバート・ハモンドとダイアン・ウォーレン (**trans**: Albert Hammond and Diane Warren) [スターシップ, Starship] |
| annotation error | マクドナルドの最初の店舗はどこ？ (**trans**: where was the very first mcdonald's built?) | 最初のマクドナルドはアメリカ合衆国・カリフォルニア州サンバーナーディノでマクドナルド兄弟が1940年に始めたものである。 (**trans**: The first McDonald's was started in 1940 by the McDonald's brothers in San Bernardino, California, United States.) | アメリカ合衆国・カリフォルニア州サンバーナーディノ [アメリカ合衆国, カリフォルニア州, サンバーナーディノ, アメリカ] |

Table 18: Examples of the Japanese error cases. "**trans**" denotes the English translation.

| Sub type | Query | Prediction [gold Answer] |
|---|---|---|
| temporal dependency | 今年のスーパーボウルはどこでありますか (**trans**: where is the super bowl being played at this year) | ルイジアナ州ニューオーリンズ [アトランタ] |
| ambiguous questions | トワイライトシリーズの本を教えてください (**trans**: what are the books in the twilight series) | ステファニー・メイヤー [エクリプス/トワイライト・サーガ, エクリプス, ニュームーン] |
| inconsistency between Wikipedias | ニューヨーク州ユーティカの人口はどのくらいですか。 (**trans**: what is the population of utica new york) | 62,235人 [60635] |

Table 19: Examples of questions labeled as underspecified questions in our error analysis.

### D.4 More Qualitative Examples

**Examples errors in MKQA Japanese questions.** Table 18 shows examples of (b) a generation language error (different lang), (c) incorrect answer generation (incorrect answer), and (d) an answer annotation error (annotation error). The first example shows an error of different language where generated text is not in the target language. Such errors are prevalent in Japanese, especially when retrieved passages are written in languages with Latin script. Transliteration of foreign words into Japanese is challenging as there are multiple ways to map English words to Japanese type script (i.e., *katakana*). Future work can improve those cross-lingual generations between languages with their own type script and the ones with Latin script. In the second example, CORA answers the song writers, instead of answering who sings the song. This shows even state-of-the-art models still exploit certain (spurious) patterns or lexical overlap to the question (Sugawara et al., 2018). The final example demonstrates the annotation difficulty of covering all possible answer aliases for multilingual open QA. Although the predicted answer is semantically correct, it's not covered by the gold answer annotations in MKQA.

We also show questions that are judged as (e) underspecified questions in Table 19 in this analysis. The first two examples show a question with temporal dependency and an ambiguous question. In the final example, we found that the information about the population of Utica, New York is different in English Wikipedia (60,635) and Japanese Wikipedia (62,235), and CORA generates an answer based on the information in Japanese.

| Query | Paragraph | Prediction |
|---|---|---|
| オレンジ・イズ・ニュー・ブラックはいつ放送される？ (**trans**:when is orange is the new black on?) | 『オレンジ・イズ・ニュー・ブラック』は、2013年7月11日よりネットフリックスで配信開始されているアメリカのテレビドラマ (**trans**:"Orange is the New Black" is an American TV drama that has been available on Netflix since July 11, 2013.) | 2013年7月11日 |
| hulkを演じる役者は誰 (**trans**: who is the actor that plays the hulk?) | Bruce Banner es un personaje interpretado primero por Edward Norton y actualmente por Mark Ruffalo en la franquicia cinematográfica Marvel Cinematic Universe (MCU) basado en el personaje de Marvel Comics del mismo nombre y conocido comúnmente por su alter ego, Hulk (**trans**: Bruce Banner is a character played first by Edward Norton and currently by Mark Ruffalo in the Marvel Cinematic Universe (MCU) film franchise based on the Marvel Comics character of the same name and commonly known by his alter ego, Hulk.) | Mark Ruffalo |
| 現在の火星の気温は？ (**trans**: what's the temperature on mars right now?) | 現在の火星の表面での年平均気温は、210K以下であり (**trans**:The current average annual temperature on Mars is less than 210K.) | 210K以下 |

Table 20: Examples of `unanswerable` Japanese MKQA questions where CORA successfully finds the correct answers. The answers are validated by the authors of this paper.

**Unanswerable MKQA questions that CORA could answer.** Table 20 shows `unanswerable` Japanese MKQA questions for which CORA can successfully find correct answers from non-English languages' text. Although MKQA answers are carefully annotated by crowd workers who extensively search online knowledge sources in English, around 30% of the questions remain unanswerable. Among the valid unanswerable questions, we found that in about 20% of the *unanswerable* questions we can find correct answers by retrieving evidence passages in another language (e.g., Japanese, Spanish). This indicates the effectiveness of cross-lingual retrieval to improve answer coverage.