# OpenReview forum: "One Question Answering Model for Many Languages with Cross-lingual Dense Passage Retrieval"
_NeurIPS.cc/2021/Conference — NeurIPS 2021 Poster_

### Official Review · Reviewer_nAYP · 2021-07-10

**Rating:** 4
**Confidence:** 4

**Summary:**

This paper presents a multilingual dense passage retrieval model called mDPR to deal with the multilingual/cross-lingual IR and QA tasks. The mDPR part is firstly trained using existing labeled multilingual Q-P pairs. Then, possible passage pairs are selected from a given multilingual corpus based on the trained mDPR and used to train an answer generation model called mGEN. Next, given each question, the trained mDPR is further used to retrieve more possible passage candidates in other languages from Wikipedia corpus, where the retrieved passages that can lead to correct answers are considered as positive instances, the others as negative instances. With these expanded new question-passage pairs, the training of mDPR is performed again. The above procedure is conducted in an iterative manner. Evaluations are conducted on TyDi QA and MKQA in the open-domain QA setting. Several baselines are included and compared. The key contribution is to do DPR in the multilingual/cross-lingual setting.

**Limitations And Societal Impact:**

no potential negative societal impact.

**Main Review:**

This is a good paper that extends dense IR from monolingual to multilingual/cross-lingual. But the novelty is limited. The most important part, I think, is the data expansion part described in Section 2.2. From model perspective, there seems be have few innovation. The baselines compared in Table 1 and 2 are also weak. Why not compare with the submissions on the TyDi QA leaderboards? Why stronger multilingual pre-trained models such as XLM-R or mBART are not used and fine-tuned for comparison?

**Time Spent Reviewing:**

2 hours

---

> ### Author Response · Authors · 2021-08-07
> **Response to reviewer nAYP**
>
> We thank the reviewer for providing insightful comments. We address your concerns raised in the original review.
>
> #### **1. Novelties**
> As discussed by the reviewer qfFE,  this is the first work introducing a unified single open-domain QA system that works in many languages. Our work uses a non-trivial combination of dense passage retrieval and answer generation modules to answer multi-lingual questions.  Our main novelty is how to integrate mDPR and mGEN with scarce cross-lingual training data by incorporating external data, and we successfully train mDPR and mGEN together, leading to significant performance improvements.  Our effort is certainly inspired by prior work, but we empirically show that a direct extension of existing techniques of multilingual encoding and generation will suffer in the languages without training data in our ablation studies (Table 4). We believe our work would impact the future work on multilingual open QA.
>
> #### **2. Baselines**
>
> As mentioned above, this is the first attempt to test a unified retriever-reader system in multilingual open-domain QA, and we compare all of the top systems on the [XOR-TyDi QA leaderboard](https://nlp.cs.washington.edu/xorqa/) at the time of submission, including the state-of-the-art systems that heavily rely on production level external APIs (e.g., Google Translate). We outperformed all of these models by up to 20 F1. Nevertheless, we conduct detailed ablation studies and test multiple variants of the CORA framework, which shows simply combining a multilingual LM-based retriever and multilingual generation model cannot achieve competitive performance, as noted by review qfEF.
>
> Unlike XOR-TyDi QA and MKQA, the TyDi QA leaderboard aims at machine reading comprehension where evidence paragraphs in the target language are given a priori and the task is to answer a question based on that paragraph. Thus, it does not require any retrieval and all of the questions can be answered based on the paragraphs written in the same language as the question. Those models in the leaderboard are, therefore, not applicable in our setting.
>
> #### **3. Other language models (e.g., XLM-R, mBART)**
>
> We have tested an XLM-R based dense passage retrieval model for dense passage retrieval, and that does not give performance improvements as noted in footnote 6. We also experimented with m-BART based generator model in our preliminary experiments, resulting in lower performance than the mT5 based generator model. For this reason, we use mT5 as our base model. It should be noted that mT5 outperforms mBART or XLM-R in various tasks and can be seen as a state-of-the-art multilingual model. We will add these results to the main table in the revision.

---

> > ### Comment · Reviewer_nAYP · 2021-08-23
> > **thank you for the rebuttal**
> >
> > Thank you for the answers to my questions. However, I will not change my score and please include all the discussions and experimental results in the next version of the paper.

---

### Official Review · Reviewer_qfFE · 2021-07-14

**Rating:** 9
**Confidence:** 5

**Summary:**

This paper studies the task of multilingual Open-Domain Question Answering.
The task setting requires building a system that can answer natural language questions in any language on an open set of domains. Models are required to answer in the same language the question was asked in.
The authors tackle the task using a fairly conventional retrieve-and-read ODQA model, with the necessary modifications to make both the retriever and reader component language-agnostic, by exploiting multi-lingual pretrained models, training on available multilingual training data, and using and a clever data augmentation setup to increase multilingual training data.

The proposed solution “CORA”, operates over wikipedias in many different languages.

The retrieval component “MDPR” is a multi-lingual extension of DPR, using MBERT. This model can retrieve evidence documents across different languages, rather than having to match the language of the question. This means that a question expressed in a low-resource language can benefit from retrieving evidence from high resource languages. It also allows for a question written in one language on a cultural topic associated with a different language to retrieve from the most appropriate language’s wikipedia.

The reader “mGEN” is a generative reader, implemented using mT5, that takes as input the concatenation of the top-K retrieved passages from the retriever (which may be several different languages) and the question, and generates the answer in the question language, token by token.

The main difficulty of the proposed solution is how to train the MDPR and MGEN model components, given the limited multilingual raining data. The initial model is trained with NQ (english) and TyDi QA and Xor-TiDi QA (multilingual data).
To increase the amount of multilingual training data and language coverage,  the authors introduce a data augmentation strategy where the initial model is used to harvest potential additional data. This exploits the wikipedia “language links” resource. This iterative approach works by retrieving passages, looking up passages on the same topic in other languages using “language links”, and using the reader to see whether these can be used to generate the correct answer - if they can, they are added as additional training data for the next round. Another expansion mechanism exploits the fact that most answers are wikipedia entities, and can be translated using annotations available in the wikipedia “language links” resource.

CORA is thoroughly evaluated on TyDI QA and an evaluation-only dataset called MKQA, and performs very strongly, improving the state-of-the-art by a wide margin in almost every setting..
Thorough ablations give insights and quantify the the effects of the system, and demonstrate zero-shot question answering abilities


**Limitations And Societal Impact:**

The authors provide a note on societal impacts which summarize the main impacts well. This kind of work will likely have a positive impact on societal equality, unlocking the strong ODQA performance we see in high resource languages for lower resource languages. There are risks that CORA may provide wrong answers that might be harmful or offensive, but this is the case for monolingual systems too. The authors rightly point out that not relying on external blackbox translators allows the model to be more interpretable and debuggable.
The use of wikipedia language links is useful, but does limit the textual domains the data expansion technique can be used for to similar domains to wikipedia.


**Main Review:**

This is an empirically very strong piece of work that performs sensible (if straightforward and somewhat incremental on the modelling side) extensions of work from open-domain QA and multilingual open-domain QA. However, the result is stronger than the sum of its parts, unlocking the ability to combine information across many languages’ wikipedias, and enjoy the resultant improvements in performance. This is a really good piece of work, and well executed.

The iterative training and data expansion techniques are interesting and important contributions, which leverage the wealth of annotations and metadata available in wikipedia to improve cross-lingual QA, without requiring explicit question translation or extra annotation efforts by humans. One limitation is that this won’t always work well if the language links aren’t high quality for some languages, or in other domains where language links aren’t available (either because wikipedia isn’t a good background corpus, or because answers are not wikipedia/wikidata entities).

The experiments and ablations are thorough and detailed and the analysis is deep.
In terms of some constructive criticism, the MGen model could probably be made more powerful by using the Fusion-in-Decoder architecture, which would allow the model to scale to reading more passages per question, which has shown to greatly improve accuracy on open-domain QA.

I also wonder why the models’ performances  are so different and sometimes inconsistent between MKQA and XOR-full (comparing tables 1 and 2)  - for example, CORA scores 54.8 F1 for arabic on XOR-full, but loses a lot of performance arabic on MKQA (only 13.1 F1) . Conversely, MT+DPR performs poorly for arabic on XOR-full (7.6 F1) but  outperforms CORA on MKQA (16.0 F1).

Summary:
* Originality: This paper takes a standard ODQA approach, but converts it to a massively cross-lingual setting in a novel and very powerful way.
( Quality: the experiments are well-thought out, the results are very strong, and the ablations and analysis are execllent
* Clarity: The paper is very clear and well written
* Significance: I expect this to be a high impact paper and serve as inspiration for many followup works and tasks.




**Time Spent Reviewing:**

4

---

> ### Author Response · Authors · 2021-08-07
> **Response to Reviewer qfFE**
>
> We really appreciate your detailed review and strong support.
>
> > In terms of some constructive criticism, the MGen model could probably be made more powerful by using the Fusion-in-Decoder architecture
>
> We agree that mGEN can be further improved by using more recent models such as FiD. In this work, we focus on using a more simple generator model as a first step towards building a single unified open-domain QA system so that follow-up work can easily analyze the results and build upon our CORA framework.
>
> > Q: I also wonder why the models’ performances are so different and sometimes inconsistent between MKQA and XOR-full (comparing tables 1 and 2).
>
> MKQA questions are translated from English Natural Questions data, and all of the questions require cross-lingual retrieval and generation between Arabic and English, whereas more than half of the TyDi-XOR questions can be answered based on Arabic Wikipedia.
> With some analyses on Japanese questions (Table 6), we realized that our model often generates correct answers in English, which are penalized. We believe a better evaluation metric or future modeling improvements in the generative component would give further performance gains.

---

### Official Review · Reviewer_hSDX · 2021-07-15

**Rating:** 5
**Confidence:** 4

**Summary:**

This paper addresses the multilingual open QA task by combining a cross-lingual passage retrieval algorithm with a multilingual autoregressive generation model. The approach is thus capable of answering questions where the answer and the question are in different languages.  The authors extended existing annotated data using an iterative training approach that improves retrieval for languages with limited resources. The experimental results showed that the proposed method (CORA) outperformed the previous state of the art on multilingual open question answering benchmarks across 26 languages.

**Limitations And Societal Impact:**

I don't see potential negative social impact of this work.

**Main Review:**

This paper targets at Multilingual Open Domain QA, which is a novel task with many application scenarios. The proposed method, CORA, showed relatively good performance on 26 languages on benchmark datasets.
However, the overall idea is very similar to CRISS [1]. The CRISS approach consists of a cross-lingual retrieval module and a seq2seq module, and they are optimized alternately. Similarly, the CORA approach in this paper also consists of DPR (Dense Passage Retriever, mbert structure) and GEN(Answer Generator, mt5 structure), and they are also optimized alternately. Although CRISS [1] doesn’t target at multilingual open QA,  it has strong performance on cross-lingual sentence retrieval, which is related to multilingual open QA. The authors should compare their work with CRISS [1].
Authors claim that they introduce a new dense passage retrieval algorithm, but don’t mention any similar work about dense passage retrieval [2].
Overall, this paper combines DPR and GEN module and achieves good experimental results. However, the technical contribution of the paper is quite limited.
[1] Cross-lingual Retrieval for Iterative Self-Supervised Training
[2] Approximate Nearest Neighbor Negative Contrastive Learning for Dense Text Retrieval


**Time Spent Reviewing:**

3

---

> ### Author Response · Authors · 2021-08-07
> **Response to Reviewer hSDX**
>
> We thank the reviewer for providing a detailed review and references.
>
> ### 1. Differences from CRISS
>
> CRISS is a self-supervised pre-training approach consisting of a parallel sentence mining module and a seq2seq model to improve cross-lingual alignments, showing improvements in machine translation and sentence retrieval tasks.
> Although both CRISS and CORA iteratively train the mining component and the generation component, there are two important differences: 1) the tasks are different; our model is designed for multilingual open-retrieval QA, which introduces new challenges compared to sentence retrieval and machine translation, and 2) the training process is different, and the previously proposed iterative approach is not enough to overcome those new challenges.
>
> ####  **1-1. Task differences**
>
> There are new challenges specific to multilingual open-retrieval QA that make it difficult to apply methodologies originally developed for machine translation or sentence retrieval to multilingual open-retrieval QA.
>
> - First, the cross-lingual sentence retrieval task is a sentence-to-sentence retrieval between predefined language pairs, while open-domain QA requires retrieving passages for questions, without knowing in which languages we can find sufficient evidence.
>
> - Second, in answer generation, the retrieved passage can be in any language included in the $\mathcal{C}^{multi}$, so our systems need to learn to generate answers from diverse multilingual evidence passages, which may not be covered by the original training data. This is unlike machine translation where the source and target languages are often specified.
>
> - Third, in the evaluation of the cross-lingual sentence retrieval (e.g., [Tatoeba](https://raw.githubusercontent.com/facebookresearch/LASER/master/data/tatoeba/v1/tatoeba.afr-eng.afr)), the retrieval pool is much smaller than open-domain QA (a few thousand sentences vs. tens of millions of passages, each of which includes hundreds of tokens). This makes cross-lingual sentence retrieval (or sentence matching) easier than cross-lingual passage retrieval for open-domain QA, where we need to deal with millions of passages.
>
> ####  **1-2. Difference in training**
> To address those important challenges related to multilingual open-retrieval QA (listed above), we use iterative training and data expansion approaches guided by mGEN filtering (explained in Section 2.2).
> In particular, to address the First and Second challenges, our model uses a trained mDPR to retrieve passages related to a question, uses external Wikipedia links to increase the diversity of cross-lingual data, and uses trained mGEN to filter out spurious passages that are not directly related to a given question. To address the third challenge related to scalability, our model obtains dense passage encodings through positive and negative mining that makes training feasible.
>
> Adding Wikipedia link-based data is particularly useful for MKQA, where many of the languages are not covered by existing training data. mGEN-based filtering is important in open-domain QA tasks, where models are known to suffer from spurious passages (passages with answer strings lacking sufficient evidence to answer). We will also add ablation results of the model trained without having language link-based data expansions and data filtering during training to illustrate those challenges.
>
> ### 2. References
>
> We will cite those recent open-domain QA papers in the related work section. Please note that our work introduces a new approach for multilingual open-retrieval QA, presenting orthogonal contributions of recent improvements on DPR in terms of performance and efficiency in the English open-domain QA task. Those approaches can be incorporated into our framework.

---

> ### Comment · Reviewer_hSDX · 2021-08-23
> **Response to rebuttal**
>
> I have read through the authors' response, and I would still hold the previous score. The authors also agree the idea is similar to CRISS. I don't think extend the method to a new application is a significant contribution.

---

> > ### Author Response · Authors · 2021-08-25
> > **Thank you for involving the discussion. We politely disagree that this work is similar to CRISS (our response, Section 1).**
> >
> > Thank you for involving the discussion.
> > As listed in our author response, we politely disagree that this work is similar to CRISS and is a simple extension of it. Again, the training schema is significantly different from CRISS (our response, 1-2. Difference in training), which is necessary to achieve many-to-many cross-lingual retrieval over millions of documents. As acknowledged by reviewers XaUF and qfFE, multilingual open-retrieval question answering is a new task that introduces new challenges beyond existing cross-lingual sentence matching or machine translation (our response, 1-1. task differences).

---

### Official Review · Reviewer_XaUF · 2021-07-16

**Rating:** 6
**Confidence:** 4

**Summary:**

The paper attempts to provide a solution to an interesting problem of cross language retrieval based question answering in a zero shot (without requirement of any language-specific annotated training data). Authors propose a new DPR algorithm that to be able to retrieve documents relevant to the question across languages (for multiple languages). The paper extends retrieve-the-generate based open domain QA system to a cross-lingual setting.

**Limitations And Societal Impact:**

Limitations of the work has not been discussed by the authors. Authors did promise to make the code publicly available to general public to enable further research in the direction ( societal impact)

**Main Review:**

How does the proposed approach differ in terms of techniques with something already proposed like RAG (https://arxiv.org/abs/2005.11401). It looks like the proposed approach is just a multilingual version of RAG thus the paper has limited novelty.  Does the system expects the  gold document annotations to be available for training the retriever module? Section 2.2., talks about positive passages, this requires huge time and effort for annotation, so how does authors claim in abstract that the proposed technique does not require any language specific annotated training data?  What happens in the scenario where a wikipedia page about a specific entity is not available in a particular language in which the question has been asked?  Section 2.2 needs to have some more intuition and clarity. What is intuition behind translating English answers to target language. Is there a mechanism to filter out spurious passages by the automatic passage labeling technique? How does the model learns cross lingual embeddings? do authors use a pre-trained model like m-bert? How does the proposed model results compare when compared with m-bert based methods?

Update: I think the problem formulation is novel. So, after reading author responses I am increasing the score to 6.

**Time Spent Reviewing:**

2

---

> ### Author Response · Authors · 2021-08-07
> **Response to Reviewer XaUF**
>
> We appreciate your insightful comments. We address your questions raised in the original review.
>
> > Q1: How does the proposed approach differ in terms of techniques with something already proposed like RAG? It looks like the proposed approach is just a multilingual version of RAG thus the paper has limited novelty.
>
> We respectfully disagree with the reviewer that our approach is just a multilingual version of RAG. RAG is an English open-domain QA system that combines a dense passage retriever with an answer generator, trained on English question/passage pairs of data.
>
> 1. A direct extension of RAG to multilingual settings is **mDPR$_1$+mGEN$_1$** in Table 4, which shows a significant drop in performance compared to our CORA. An important reason for this drop is the lack of the cross-lingual annotated training data in most of the non-English languages (as discussed in L97-101 and also mentioned by reviewer qfFE). This prevents one from building a single unified open-retrieval QA system that retrieves documents cross-lingually. On the other hand, our system--CORA--uses an iterative data expansion and training approach to train a model that is capable of cross-lingual dense passage retrieval.
>
> 2. Here is how we have solved the challenge: we first fine-tune mDPR and mGEN using existing data. Then, we mine new positive and negative passages by retrieving top passages from large document collections using the trained mDPR. We additionally retrieve non-English articles corresponding to the original English articles annotated in Natural Question data. We then run mGEN on a pair of a question $q_i$ and a passage $p_{ij}$ and see if mGEN can generate the correct answer given the passage. If mGEN succeeds, we label $p_{ij}$ positive otherwise negative, to filter out false-positive passages. The new positive and negative passages will be used to fine-tune mDPR. We repeat this process multiple times.
>
>
> > Q2: Does the system expect the gold document annotations to be available for training the retriever module? ... how does authors claim in abstract that the proposed technique does not require any language specific annotated training data?
>
> Our model uses gold document annotations for some languages (8 languages available in XOR-TyDi QA and Natural Questions) and shows how to extend “even to languages where we do not have any QA training data” for retrieval or generation (20 more languages). In particular, we use existing TyDi QA, XOR-TyDi QA, and Natural Questions data, which only covers 8 languages among 28 languages tested in the paper. By iteratively training mDPR and mGEN with Wikipedia language-link-based data expansion, our system can automatically mine new training data without new human annotation or using external modules such as machine translations.
> We make sure to clarify this in the abstract and intro.
>
> > Q3: What happens in the scenario where a Wikipedia page about a specific entity is not available in a particular language in which the question has been asked? Section 2.2 needs to have some more intuition and clarity.
>
> We have empirically shown that even in the languages where we do not have any Wikipedia passages or training data such as Malay, our model can perform well (around 30 F1). In contrast, prior approaches such as Translate-test and BM 25 will suffer in those languages if applicable. Please see the full results in Tables 2 and 3. This is mainly because for a question in a language with no Wikipedia resource, our mDPR can retrieve passages among our multilingual collection (in a different language from the original question). Moreover, our mGEN can generate answers to the original question given a collection of passages in different languages. To train mDPR and mGEN to learn cross-lingual retrieval and generation, we introduce iterative training and data expansion schema discussed in detail in the responses to Q1 and Q4.
>
> > Q4: What is intuition behind translating English answers to target language.
>
> We use this step for data expansion (Section 2.2; L110-111 and L120-126).
> Cross-lingual answer generation from scare training data is challenging and only using multilingual generation models such as mT5 is not enough. We automatically translate the English answers into the target languages and create cross-lingual data synthetically to train mGEN. Without using those data, we observe large performance deterioration in MKQA, where 22 languages do not have any human annotated QA train data. In our updated version, we will include ablation results without synthetic data training, and also provide detailed motivations in L110-111.
>
> > Q5: Is there a mechanism to filter out spurious passages by the automatic passage labeling technique?
>
> As mentioned in L127-133, we run mGEN on each passage to see if mGEN can generate correct answers given the passage. We assume that when mGEN fails to generate a correct answer given the passage, the passage may not provide sufficient evidence to answer so that it helps to filter out spurious passages. We will discuss the motivation in more detail in the updated version.
>
> > Q7: do authors use a pre-trained model like m-bert? How does the proposed model results compare when compared with m-bert based methods?
>
> Our mDPR model is based on the multilingual BERT model. We also compare our model with multilingual BERT-based DPR which has been trained on Natural Question data only or initial training data (DPR (NQ), Tables 4 and  5), and multilingual BERT-based DPR shows significantly lower performance than CORA in terms of both retrieval and final QA performance.
>
> > Q6: How does the model learn cross lingual embeddings?
>
> As mentioned in L230-231, our cross-lingual iterative training enables our model to locate the questions embeddings close to the relevant passages embeddings in different languages. The key intuition is that our augmentation adds external data, whereas without such steps only scarce cross-lingual retrieval is available (See Appendix Fig. 5 for t-SNE visualization).  As shown in Figure 3, after the iterative training, our mDPR successfully retrieves documents from multilingual Wikipedia passages, which leads to significant performance improvements from prior work or our competitive baselines.
>
> > Limitations of the work have not been discussed by the authors.
>
> We have discussed the potential limitations of CORA in our analysis section (L229-232, L243-253) and the Social impact section (L287-292) as also noted by reviewer qfEF. We found that although CORA performs really well in many zero-shot unseen languages, it still suffers particularly in low-resource languages such as Khmer. We also found that in languages using their own scripts (e.g., Japanese), CORA sometimes generates semantically correct answers in other languages (e.g., English). Without relying upon external black-box APIs as in prior work, we can easily analyze the model’s behavior, which helps us to improve the model to tackle those technical challenges. Again, this is the first work to propose a single unified open-domain QA system in many languages without relying on machine translation and has shown significantly large performance improvements from prior approaches. We will create a separate section to summarize those aforementioned discussions in the revision.

---

> > ### Comment · Reviewer_XaUF · 2021-08-23
> > **Thanks for the rebuttal. Increasing score to 6**
> >
> > I have read your response to all the points, I agree that the problem formulation is novel (Multilingual ODQA with retrieval) and the proposed solution is not just direct extension of RAG. So, I am increasing my score to 6.

---

> > > ### Author Response · Authors · 2021-08-25
> > > **Thank you for  involving the discussion and acknowledging the novelties.**
> > >
> > > Thank you so much for reading our response and acknowledging the novelties of the task formulation and proposed method.

---

### Decision · Program_Chairs · 2021-09-27

**Decision:**

Accept (Poster)

**Comment:**

The paper tackles the problem of multilingual open domain question answering, introducing an iterative approach to mining answer passages.

There is not a clear consensus amongst the reviewers. It is clear that this is an interesting and under-explored problem, and that the proposed solution works well. Reviewer qfFE argues strongly for acceptance based on the novelty of the setting, the strong empirical comparisons, and the potential for inspiring future work.

The major criticism made of the paper is its lack of technical novelty, and reviewers agree that similar iterative retrieval ideas have been proposed in models such as RAG and CRISS. However, there is a non-trivial modeling contribution over these papers to extend them successfully to multilingual open domain QA. Reviewer nAYP also feels that the baselines are weak, but I agree with reviewer qfFE that the strong set of baselines is a significant contribution of the work.

Overall, I recommend acceptance.